# A multi collocation method for coastal zone observations with applications to SENTINEL-3a altimeter wave height data

Schulz-Stellenfleth Johannes and Staneva Joanna

Helmholtz Zentrum Geesthacht (HZG), Institute of Coastal Research (IfK)

Max-Planck-Str. 1, 21502 Geesthacht, Germany

**Correspondence:** Johannes Schulz-Stellenfleth (johannes.schulz-stellenfleth@hzg.de)

**Abstract.** In many coastal areas there is an increasing number and variety of observation data available, which are often very heterogeneous in their temporal and spatial sampling characteristics. With the advent of new systems, like the radar altimeter onboard the SENTINEL-3a satellite, a lot of questions arise concerning the accuracy and added value of different instruments and numerical models. Quantification of errors is a key factor for applications, like data assimilation and forecast improvement.

In the past, the *triple collocation* method to estimate systematic and stochastic errors of measurements and numerical models was successfully applied to different data sets. This method relies on the assumption, that three independent data sets provide estimates of the same quantity. In coastal areas with strong gradients even small distances between measurements can lead to larger differences and this assumption can become critical. In this study the *triple collocation* method is extended in different ways with the specific problems of the coast in mind. In addition to nearest neighbor approximations considered so far, the

presented method allows use of a large variety of interpolation approaches to take spatial variations in the observed area into account. Observation and numerical model errors can therefore be estimated, even if the distance between the different data sources is too big to assume that they measure the same quantity. If the number of observations is sufficient, the method can also be used to estimate error correlations between certain data source components. As a second novelty, an estimator for the uncertainty of the derived observation errors is derived as a function of the covariance matrices of the input data and the number

of available samples.

In the first step, the method is assessed using synthetic observations and Monte Carlo simulations. The technique is then applied to a data set of SENTINEL-3a altimeter measurements, insitu wave observations, and numerical wave model data with a focus on the North Sea. Stochastic observation errors for the significant wave height, as well as bias and calibration errors are derived for the model and the altimeter. The analysis indicates a slight overestimation of altimeter wave heights, which

becomes more pronounced at higher sea states. The smallest stochastic errors are found for the insitu measurements.

Different observation geometries of insitu data and altimeter tracks are furthermore analysed, considering 1D and 2D interpolation approaches. For example, the geometry of an altimeter track passing between two insitu wave instruments is considered with model data being available at the insitu locations. It is shown, that for a sufficiently large sample, the errors of all data sources, as well as the error correlations of the model, can be estimated with the new method.

## 1 Introduction

Coastal areas like the German Bight are often characterised by strongly heterogeneous ocean dynamics, typically associated with complicated bathymetry, small scale coastline features, and river runoffs. A few instruments, like HF radar are able to capture at least 2D surface currents with large coverage and high resolution quite nicely. Such systems have a typical range of about 100 km, spatial resolutions on a kilometre scale and about 20 min sampling (Stanev et al., 2015). However, most instruments, provide only point measurements (e.g., buoys), or transects (e.g., satellite altimeter). The combination and interpretation of such data is therefore often a challenge. In heterogeneous coastal areas with strong gradients, spatially distributed instruments can observe very different components of the dynamics, even if they are in close proximity.

In the following, this situation is studied in more detail with respect to ocean waves and the significant wave height in particular. Wave height information is of paramount importance for many applications, e.g., shipping, offshore operations, or coastal protection. Although numerical wave forecast models have reached an impressive level of accuracy, there is still room for improvement, in particular in coastal areas with complicated dissipation processes associated with wave breaking and bed friction (Woolf et al., 2002; Reistad et al., 2011; Voorrips et al., 1997; Herbers et al., 2000; Bouws and Komen, 1983; Young et al., 2013; Semedo et al., 2015), as well as with coupling processes between ocean waves, ocean circulation and the atmosphere (Cavaleri et al., 2018; Staneva et al., 2017; Alari et al., 2016). The focus in this study is on the North Sea, which has an interesting ocean wave dynamics mainly caused by the semi-enclosed geometry (Semedo et al., 2015; Voorrips et al., 1997; Boukhanovsky et al., 2007; Staneva et al., 2014). The bathymetry of the considered area with the locations of some insitu wave measurement stations used in the following analysis is shown in Fig. 1.

Traditionally, validations of new data sets are performed by comparing to data from established standard insitu measurements, which are regarded as a reference. As a first step this is acceptable, however one has to take into account that these reference instruments are affected by measurement errors as well, and the separation of the error contributions from the new data set and the reference instrument is, in general, not possible unless additional information is used. This is easy to see, if two data sets $x$ and $y$ with uncorrelated additive noise are considered, i.e.,

$$x = t + \epsilon_x \tag{1}$$
$$y = t + \epsilon_y . \tag{2}$$

where $t$ represents the "truth". If statistics is performed on the difference $\xi$ of $x$ and $y$, one gets for the mean squared error

$$\xi = \langle (x - y)^2 \rangle = \langle \epsilon_x^2 \rangle + \langle \epsilon_y^2 \rangle , \tag{3}$$

and it is apparent, that it is not possible to derive either $\langle \epsilon_x^2 \rangle$ or $\langle \epsilon_y^2 \rangle$ from $\xi$ alone. The usual approach is therefore, to use additional data sets and to make certain *a priori* assumptions about the errors. If only one data set is added, this leads to the *triple collocation* method, which has been used and discussed in a number of previous studies (Janssen et al., 2007; Vogelzang

and Stoffelen, 2012; Stoffelen, 1998; Caires and Sterl, 2003; McColl et al., 2014). Collocation studies, as presented here, often use a mixture of observations and numerical models. The term "data source" will therefore be used in the following to refer to different types of input data.

In this study the *triple collocation* approach is extended and adjusted with the special requirements of the coast in mind, where one can usually expect stronger gradients and smaller scale variations than in the open ocean. The objective of the study is to deal with the following four specific issues:

- In the *triple collocation* method, different information sources within a certain distance are assumed to measure the same quantity, which can be unrealistic in regions with strong gradients, like most coastal areas.

- So far, assumptions about correlations errors were made *a priori* (Vogelzang and Stoffelen, 2012), but they were not obtained as a result of the collocation process.

- So far, no systematic approach was presented to deal with more than 3 data sources.

- The quantification of uncertainties concerning estimations of systematic and stochastic data source errors was so far only done based on boot strap approaches (Caires and Sterl, 2003).

The question about the accuracy of error estimates is of particular concern for new instruments, like SENTINEL-3a, for which the amount of available data is still relatively limited. It is also clear, that collocation distances are of concern mainly for point measurements or transect observations from satellites. The interpolation of numerical model data to given observation locations is usually less critical, if the spatial resolution is appropriate.

The work presented here addresses the issues mentioned above and makes the following main contributions:

- A generalisation of the *triple collocation* method is introduced, where the "truth" is not necessarily represented by a single number, but by a more general parameterisation of the truth state, that is measured by a group of instruments within a certain distance. The analysis presented here concentrates on 1D models (i.e., lines), and 2D models (i.e., planes), but can be easily extended to include more sophisticated approaches.

- In certain configurations, i.e., definitions of "truth" vectors and spatial distributions of data sources, the approach allows an estimation of cross covariance components of the stochastic errors contained in the considered observations or numerical models.

- The theory includes the definition of a general data source vector, which can contain an arbitrary number of observations and numerical model data.

- Analytical expressions are derived for the estimation errors regarding both systematic calibration errors and stochastic errors of the different data sources.

Like the standard *triple collocation* method, the extended approach also provides estimates of systematic bias and calibration errors. We will refer to the standard *triple collocation* method as "TRIPCOL", and to the multi collocation as "MULTCOL" in the following.

As an example for the generalised parameterisation of the "truth", one can imagine two wave buoys and a satellite altimeter track passing between the two of them. Lets furthermore think about a situation, where the wave buoys are too far away from the track to assume that all three instruments measure the same quantity. However, it maybe an acceptable assumption, that the wave height measured by the altimeter is a linear combination of the wave heights observed by the two buoys. If independent numerical model wave height estimates are available at the buoy locations, the method presented in the following provides a systematic approach to estimate not only the stochastic errors of all data sets, but also the error correlation of the model at the buoy locations.

The present study is supposed to make a contribution to the exploitation of measurements with larger distances, where additional assumptions about the spatial variation of the "truth" are required. As an illustration, Fig. 2 shows maps of the North Sea with altimeter tracks and collocated buoys with the color coding referring to the number of obtained collocated data samples within the period April 2016 to August 2017. The data sets will be introduced in more detail in Section 3. The plot Fig. 2a shows the situation, if a collocation distance of 10 km is assumed as acceptable, whereas Fig. 2b shows the same with a collocation distance of 20 km. One can see, that the number of data sets increases rapidly if larger distances are considered.

With regard to the estimation errors, expressions are derived, which provide a quantification depending on the covariance matrices of the data sources, and the number of available data samples. These results can give valuable information on the trustworthiness of estimated observation errors, in particular in situations with a small number of samples.

The paper is structured as follows: The multi collocation method is introduced in Section 2. This includes the explanation of the underlying theory for the treatment of the stochastic and systematic errors in Sections 2.1 and 2.2, as well Monte Carlo simulations to illustrate and verify the method. In Section 3 the analysed significant wave heights from insitu stations, SENTINEL-3a altimeter, and numerical model wave height data are introduced. As a special case of the multi collocation method, the *triple collocation* technique is applied to the wave height data sets in Section 4. This includes a new step in the analysis, in which estimation errors are quantified. Section 5 describes the combination of more than three observations taken at a certain distance to estimate measurement errors and error correlations.

## 2   Multi collocation method

In this section the multi collocation method is explained, which includes the *triple collocation* technique as a special case. In the first step, the approach for the estimation of the stochastic errors is presented, and in the second part systematic bias and calibration errors are considered.

### 2.1   Symmetric Approach

The approach presented in this section to estimate stochastic errors does not require bias-free reference instruments. Calibration errors are not considered in this first step. Lets assume the "truth" is given by a vector $\mathbf{t}$ of dimension $n_t$, and $n_o$ data sources $y_1, \ldots, y_{n_o}$ are related to the "truth" by

$$\mathbf{y} = A\mathbf{t} + \epsilon + \mathbf{b} \,. \tag{4}$$

Here, $A$ is an $n_o \times n_t$ matrix, $\epsilon$ is an $n_o$-dimensional zero mean Gaussian process, which represents the stochastic data source errors, and $\mathbf{b}$ is a vector of length $n_o$ containing the biases of the different data source components. Bold typing is used for vectors. The *triple collocation* method is then a spacial case with $n_t = 1$, $n_o = 3$, and $A = (1,1,1)^T$. Here and in the following, the symbol $T$ denotes the transpose operation. This case will be considered in Section 4 looking at a larger number of insitu observation locations in the North Sea. Using different definitions of the "truth" vector and the matrix $A$, various relationships between the "truth" and the data sources can be formulated with the above approach. In this study, we will concentrate on 1D and 2D linear models. It should be emphasized, that the "truth" cannot, in general, be represented by a finite number of parameters. However, it is reasonable to assume, that the reality is sufficiently smooth, and hence a Taylor expansion can be applied. The *triple collocation* method is then a special case, where only the constant term is considered. Depending on the number of available observations, the approach in eq. 4 allows the addition of higher order terms. We will concentrate on linear approximations in this study, however the method is able to deal with interpolation approaches of higher order, if a sufficient number of data sources is available. Conceptually, this issue is related to the topic of representation errors (e.g., Van Leeuwen (2015)). The 1D case will be considered in the Monte Carlo simulations presented in Section 2.5, as well as in Section 5.1. The 2D case will be discussed in Section 5.2.

Lets now define a matrix $B$, which contains a basis of the null-space of $A$ as rows. This can, for example, be obtained by singular value decomposition of $A$ and selecting the eigenvectors corresponding to vanishing eigenvalues. If $A$ has full rank, $B$ is a $(n_o - n_t) \times n_o$ matrix. For the *triple collocation* method this leads to

$$B = \frac{1}{\sqrt{2}} \begin{pmatrix} 1 & -1 & 0 \\ 1 & 0 & -1 \end{pmatrix} . \tag{5}$$

Multiplying eq. 4 from the left by $B$ gives

$$B\mathbf{y} = B\epsilon + B\mathbf{b} . \tag{6}$$

Averaging over all measurements then leads to

$$\langle B\mathbf{y} \rangle = B\mathbf{b} . \tag{7}$$

Forming the second order moments results in

$$\begin{aligned} B\langle \mathbf{y}\mathbf{y}^T \rangle B^T - \langle B\mathbf{y} \rangle \langle B\mathbf{y} \rangle^T &= BA\langle \mathbf{t}\mathbf{t}^T \rangle A^T B^T + B\langle \epsilon\epsilon^T \rangle B^T \\ &= B\langle \epsilon\epsilon^T \rangle B^T =: Z , \end{aligned} \tag{8}$$

where we have a symmetric $(n_o - n_t) \times (n_o - n_t)$ matrix on both sides of the equation. Because of the symmetry, one gets

$$m = \frac{(n_o - n_t)^2 + (n_o - n_t)}{2} \tag{9}$$

equations. The right hand side $Z$ is of the form

$$Z_{ij} \;\; = \;\; \sum_{q,k=1}^{n_o} \langle \epsilon_q \epsilon_k \rangle B_{iq} B_{jk} \tag{10}$$

$$= \;\; \sum_{k=1}^{n_o} \langle |\epsilon_k|^2 \rangle B_{ik} B_{jk} + \sum_{q<k} \langle \epsilon_q \epsilon_k \rangle (B_{iq} B_{jk} + B_{ik} B_{jq}) \, . \tag{11}$$

Eq. 8 is therefore a linear system of equations of the form

$\quad \mathbf{r} = D \bar{\epsilon} \, , \tag{12}$

where the vector $\bar{\epsilon}$ contains the unknown variances and covariances of $\epsilon$ and $\mathbf{r}$ contains elements of the matrix on the left hand side of eq. 8. If it is possible to limit the number of unknowns to $m$, or less, using appropriate assumptions about the variance structure (e.g., independence of error components), this system, can be solved, if the corresponding system matrix $D$ is regular. Table 1 summarises some feasible combinations of $n_t$, $n_o$, and the number of error variances $n_{var}$ and covariances $n_{covar}$, that

can be estimated, if $D$ is regular. Possible observation system configurations corresponding to these cases are shown in Fig. 3. Here, Fig. 3a corresponds to the standard TRIPCOL approach, where all data sources within a certain distance are assumed to measure the same "truth". Linear approximations in 1D and 2D used in the MULTCOL approach, to relate data sources with a larger distance, are depicted in Fig. 3b and 3c respectively.

If there are more equations than unknowns, a standard linear squares approach can be used to find a reasonable estimate

for the unknown variance and covariance components of $\epsilon$. It is interesting to note, that this approach also works for biased measurements, although it is in general not possible to estimate the bias explicitly. All that is required, is an estimate of $B\mathbf{b}$ and this is easy to obtain via averaging of eq. 7.

For the case of the *triple collocation* method, the system matrix $D$ is in fact regular and the inverse is given by

$$D^{-1} = 2 \begin{pmatrix} 0 & 0 & 1 \\ 1 & 0 & -1 \\ 0 & 1 & -1 \end{pmatrix} \, . \tag{13}$$

For the *triple collocation* problem this leads to the well know expressions for the stochastics error variances (Janssen et al., 2007).

$$\langle \epsilon_1^2 \rangle \;\; = \;\; \langle (y_1 - y_2)(y_1 - y_3) \rangle \tag{14}$$

$$\langle \epsilon_2^2 \rangle \;\; = \;\; \langle (y_2 - y_1)(y_2 - y_3) \rangle \tag{15}$$

$$\langle \epsilon_3^2 \rangle \;\; = \;\; \langle (y_3 - y_2)(y_3 - y_1) \rangle \, . \tag{16}$$

This corresponds to the "0D" case in Table 1 and the geometry in Fig. 3a.

If the available number of samples $n_s$ is small, the estimated observation errors maybe affected by large errors. To quantify these uncertainties at least in an approximate way, the covariance of the covariance estimator

$$\overline{\mathrm{COVAR}}(x_i, x_j) = \frac{1}{n_s} \sum_{q=1}^{n_s} x_i^q x_j^q \tag{17}$$

is considered, where the stochastic vector $(x_1, x_2, \ldots)$ is assumed to be Gaussian and zero mean. The covariance of these estimators $\chi_{i,j,i',j'}$ for different pairs of $(i,j)$ and $(i',j')$ can then be written as

$$\chi_{i,j,i',j'} := \mathrm{COVAR}(\overline{\mathrm{COVAR}}(x_i,x_j), \overline{\mathrm{COVAR}}(x_{i'} x_{j'})) = \frac{1}{n_s^2} \sum_{qq'} \langle x_i^q x_j^q x_{i'}^{q'} x_{j'}^{q'} \rangle - \frac{1}{n_s^2} \sum_{qq'} \langle x_i^q x_j^q \rangle \langle x_{i'}^{q'} x_{j'}^{q'} \rangle \,. \tag{18}$$

Using standard relationships for the higher order central moments of Gaussian distributed variables (Triantafyllopoulos, 2003), this can be expressed as

$$\chi_{i,j,i',j'} = \frac{1}{n_s} \mathrm{COVAR}(x_i, x_{i'}) \, \mathrm{COVAR}(x_j, x_{j'}) + \frac{1}{n_s} \mathrm{COVAR}(x_i, x_{j'}) \, \mathrm{COVAR}(x_j, x_{i'}) \,. \tag{19}$$

The latter expression for $\chi_{i,j,i',j'}$ can be used to estimate the variances and covariances of the estimation errors on the left hand side of eq. 8 and eq. 12 respectively. Therefore, the uncertainties of the estimated vector $\bar{\epsilon}$ can be approximated by

$$\mathrm{covar}(\bar{\epsilon}) = D^{-1} \mathrm{covar}(\mathbf{r})(D^{-1})^T \,. \tag{20}$$

From eq. 19 and eq. 20 it is evident, that observations with large variance and strong positive correlations will tend to lead to stronger estimation errors for $\bar{\epsilon}$. This is in particular the case, when the geophysical background statistics already contributes a lot of variance, or when measurements are within the correlation distance of the background fields and the uncorrelated observation errors are relatively small. The usefulness of the approximation eq. 20 will be considered in Section 2.3 based on Monte Carlo simulations.

## 2.2 Use of reference instruments

In this section a more special, but also typical situation is considered, where for a couple of measurements systematic errors can be neglected. Typically, this assumption is made for standard insitu observations systems, like wave rider buoys (Janssen et al., 2007), or wind anemometers (Stoffelen, 1998). In this case, the error model for the different data sources can be formulated as follows:

$$\begin{pmatrix} \mathbf{x} \\ \mathbf{y} \end{pmatrix} = \begin{pmatrix} I \\ \lambda \end{pmatrix} \begin{pmatrix} A_x \\ A_y \end{pmatrix} t + \begin{pmatrix} \epsilon_x \\ \epsilon_y \end{pmatrix} + \begin{pmatrix} 0 \\ \mathbf{b_y} \end{pmatrix} \tag{21}$$

Here, $\mathbf{x}$ represents the vector of reference measurements, and $\mathbf{y}$ contains the remaining data sources. In the examples discussed in the following sections, $\mathbf{x}$ will contain insitu wave height measurements, and $\mathbf{y}$ will represent a combination of satellite altimeter and numerical wave model data. The dimensions of $\mathbf{x}$ and $\mathbf{y}$ are denoted by $n_x$ and $n_y$ in the following. The matrices $A_x$ and $A_y$ translate the truth vector $\mathbf{t}$ to the expected reference measurements $\mathbf{x}$ and the other data sources $\mathbf{y}$. In addition, it is assumed that the matrix $A_x$ is invertible, i.e., it is possible to obtain an estimate of the truth vector $\mathbf{t}$ from the observations $\mathbf{x}$. The matrix $I$ is the identity matrix. Apart from a possible bias, the vector of data sources $\mathbf{y}$ maybe also affected by systematic calibration errors represented by the diagonal matrix $\lambda$.

To obtain expressions for the scaling parameters contained in $\lambda$, the first and second order moments of the input data $\mathbf{x}$ and $\mathbf{y}$ are considered. For the first order moments $M_x$, $M_y$ of $x$ and $y$, one gets

$$\mathbf{M_x} = A_x \langle \mathbf{t} \rangle \tag{22}$$

$$\mathbf{M_y} = \lambda A_y \langle \mathbf{t} \rangle + \mathbf{b_y} = \lambda A_y A_x^{-1} M_x + \mathbf{b_y} . \tag{23}$$

The second order moments $M_{xx}$ and $M_{yy}$ follow as

$$M_{xx} = A_x \langle \mathbf{tt}^T \rangle A_x^T + \langle \epsilon_x \epsilon_x^T \rangle \tag{24}$$

$$M_{yy} = \lambda A_y \langle \mathbf{tt}^T \rangle A_y^T \lambda + \langle \epsilon_y \epsilon_y^T \rangle + \mathbf{b_y} \mathbf{b_y}^T + \lambda A_y \langle \mathbf{t} \rangle \mathbf{b_y}^T + \mathbf{b_y} \langle \mathbf{t} \rangle A_y^T \lambda . \tag{25}$$

The covariance functions $C_{xx}$ and $C_{yy}$ can then be written as

$$C_{xx} = A_x \langle \mathbf{tt}^T \rangle A_x^T + \langle \epsilon_x \epsilon_x^T \rangle - A_x \langle \mathbf{t} \rangle \langle \mathbf{t} \rangle^T A_x^T \tag{26}$$

$$C_{yy} = \lambda A_y \langle \mathbf{tt}^T \rangle A_y^T \lambda + \langle \epsilon_y \epsilon_y^T \rangle + \mathbf{b_y} \mathbf{b_y}^T + \lambda A_y \langle \mathbf{t} \rangle \mathbf{b_y}^T + \mathbf{b_y} \langle \mathbf{t} \rangle^T A_y^T \lambda - (\lambda A_y A_x^{-1} \mathbf{M_x} + \mathbf{b_y})(\lambda A_y A_x^{-1} \mathbf{M_x} + \mathbf{b_y})^T \tag{27}$$

$$= \lambda A_y \langle \mathbf{tt}^T \rangle A_y^T \lambda - \lambda A_y A_x^{-1} \mathbf{M_x} \mathbf{M_x}^T (A_x^{-1})^T A_y^T \lambda + \langle \epsilon_y \epsilon_y^T \rangle \tag{28}$$

and the cross covariance $C_{xy}$ between $\mathbf{x}$ and $\mathbf{y}$ as

$$C_{xy} = A_x \langle \mathbf{tt}^T \rangle A_y^T \lambda + \langle \epsilon_x \epsilon_y^T \rangle - A_x \langle \mathbf{t} \rangle \langle \mathbf{t} \rangle^T A_y^T \lambda \tag{29}$$

$$\lambda A_y A_x^{-1} C_{xy} = \lambda A_y \langle \mathbf{tt}^T \rangle A_y^T \lambda - \lambda A_y A_x^{-1} M_x M_x^T (A_x^{-1})^T A_y^T \lambda . \tag{30}$$

The equation for $C_{yy}$ then gives

$$C_{yy} = \lambda A_y A_x^{-1} C_{xy} + \langle \epsilon_y \epsilon_y^T \rangle . \tag{31}$$

This results in $n_y$ equations for each scaling component according to

$$\lambda_i = \frac{C_{ij} - \langle \epsilon_i \epsilon_j \rangle}{\sum_q \nu_{iq} C_{qj}} := \frac{\Omega_1}{\Omega_2} \quad j = 1, \ldots, n_y \tag{32}$$

where $\nu_{iq}$ are the elements of the matrix $A_y A_x^{-1}$.

Lets assume for a moment, that the scaling parameters $\lambda$ are available. One can then derive the bias of $\mathbf{y}$ from eq. 23. Furthermore, defining the matrix $A$ in eq. 4 as

$$A = \begin{pmatrix} A_x \\ \lambda A_y \end{pmatrix} , \tag{33}$$

the approach in Section 2.1 can be applied to estimate the stochastic errors of the different data sources.

There are now two basic approaches to estimate the scaling parameters:

– Direct method: Those equations in eqs. 32 are used, for which $\langle \epsilon_i \epsilon_j \rangle$ is known, e.g., because the error components are assumed independent. In this case the estimation of the observation errors and the scaling parameters are independent and can be treated separately.

– Iterative method: Equations in eqs. 32 are used, for which $\langle \epsilon_i \epsilon_j \rangle$ is not known a priori. In this case an iterative method has to be used, where the estimation of the data source errors and the scaling parameters are performed in succession until convergence is achieved. Similar iteration techniques were also discussed for the *triple collocation* method in Janssen et al. (2007) and Vogelzang and Stoffelen (2012).

In Janssen et al. (2007) an iterative method had to be applied for the *triple collocation* analysis, because the proposed procedure for the scaling parameter estimation lead to a nonlinear expression, which could not be treated in a direct way. The direct method for the standard *triple collocation* problem leads to the known expressions (Caires and Sterl, 2003):

$$\lambda_{y_1} = \frac{C_{y_1 y_2}}{C_{x_1 y_2}} \tag{34}$$

$$\lambda_{y_2} = \frac{C_{y_1 y_2}}{C_{x_1 y_1}} \ . \tag{35}$$

One can see, that for the estimation of $\lambda_{y_1}$ no use is made of correlations between $y_1$ and $x_1$, which may contain a lot of useful information. This can be overcome by the iterative version with

$$\lambda_{y_1} = \frac{C_{y_1 y_1} - \langle |\epsilon_{y_1}|^2 \rangle}{C_{x_1 y_1}} \tag{36}$$

$$\lambda_{y_2} = \frac{C_{y_2 y_2} - \langle |\epsilon_{y_2}|^2 \rangle}{C_{x_1 y_2}} \tag{37}$$

In some cases, there maybe several equations for one component of $\lambda$, and it is then important to have an approximation for
the respective estimation errors to pick the estimator with the smallest variance. Quantification of these uncertainties is also of general interest in the statistical analysis of data, in particular if the sample size is small. For the analysis in this study, we only consider the direct method, where the $\langle \epsilon_i \epsilon_j \rangle$ in eq. 32 are known constants. We also do not consider the additional uncertainty, which is caused by estimation errors for these stochastic error variances and covariances. Denoting the nominator and denominator in eq. 32 by $\Omega_1$ and $\Omega_2$, a Taylor expansion gives

$$\lambda_k \approx \frac{1}{\langle \Omega_2 \rangle} (\Omega_1 - \langle \Omega_1 \rangle) - \frac{\langle \Omega_1 \rangle}{\langle \Omega_2 \rangle^2} (\Omega_2 - \langle \Omega_2 \rangle) + \frac{\langle \Omega_1 \rangle}{\langle \Omega_2 \rangle} \ . \tag{38}$$

For the variance one gets

$$var(\lambda_k) = \frac{var(\Omega_1)}{\langle \Omega_2 \rangle^2} + \frac{var(\Omega_2) \langle \Omega_1 \rangle^2}{\langle \Omega_2 \rangle^4} - 2 \frac{covar(\Omega_1, \Omega_2) \langle \Omega_1 \rangle}{\langle \Omega_2 \rangle^3} \ . \tag{39}$$

The variances and covariances of $\Omega_1$ and $\Omega_2$ can be derived making again use of eq. 19.

## 2.3   Generation of background statistics

In the following, the techniques presented in Sections 2.1 and 2.2 will be assessed and validated based on synthetic observations, for which the observation errors are known *a priori*. This requires Monte Carlo simulations, for which a realistic background statistics is desirable. Here, we use parameters derived from a 11-month time series of two buoys in the German Bight. The buoys "ELB" and "HEL" can be found in Fig. 1 as the instruments closest to the entrance of the river Elbe. The

buoy "HEL" is near the island Helgoland in about 25 m water depth and about 30 km north west of the buoy "ELB", which is in about 27 m water depth. The wave height distributions of both buoys shown in Figs. 4 b) and c) indicate a shape, which can be very well approximated with a log-normal distribution superimposed as green curves. The joint distribution in Fig. 4a shows a quite good correlation between the two data sets, which is expected due to the relative close proximity of the buoys.

The histogram of the difference between the Elbe buoy and the Helgoland buoy shown in Fig. 4d, indicates that the majority of cases have higher waves at the Helgoland location than the Elbe location. This makes sense, because north-westerly winds are predominant in the area. Therefore, situations with waves coming from offshore and being dissipated by wave breaking and bottom friction are most often observed in the German Bight. The fewer cases with higher waves near Helgoland are associated with southerly winds, where waves are actually generated near the coast and the wave height increases with fetch length. The

respective parameters for the log-normal distribution including the correlations of both buoy time series are given in Table 2.

## 2.4   Impact of coastal gradients and spatial data set resolutions on *triple collocation*s

In this section a brief analysis is presented concerning the impact of coastal gradients on the standard *triple collocation* approach and the role of spatial data set resolutions. The analysis is illustrated using the background statistics presented in Section 2.3.

As explained before, the *triple collocation* method makes the assumption that all three data sets represent the same "truth". We consider the case now, where this assumption is violated, and where we have data sets representing the wave height at three different locations $x, x', x''$. Lets denote the wave heights at the three locations by

$$
\begin{aligned}
H_x &= \overline{H}_x + \hat{H}_x \\
H_{x'} &= \overline{H}_{x'} + \hat{H}_{x'} \\
H_{x''} &= \overline{H}_{x''} + \hat{H}_{x''} \ ,
\end{aligned}
\tag{40}
$$

where $\overline{H}_x, \overline{H}_{x'}, \overline{H}_{x''}$ are the respective mean values, and $\hat{H}_x, \hat{H}_{x'}, \hat{H}_{x''}$ are the departures from the mean. Furthermore, it is assumed that the three wave height data sets are affected by uncorrelated additive zero mean errors $\epsilon_x, \epsilon_{x'}, \epsilon_{x''}$. According to eq. 14, the measurement error of the data source corresponding to location $x$ would be estimated as follows, if the standard *triple collocation* method is applied:

$$
\begin{aligned}
\quad \langle \epsilon_x^2 \rangle &\approx \langle (\hat{H}_x + \overline{H}_x + \epsilon_x - \hat{H}_{x'} - \overline{H}_{x'} - \epsilon_{x'})(\hat{H}_x + \overline{H}_x + \epsilon_x - \hat{H}_{x''} - \overline{H}_{x''} - \epsilon_{x''}) \rangle \\
&= \langle \epsilon_x^2 \rangle + \langle (\hat{H}_x - \hat{H}_{x'})(\hat{H}_x - \hat{H}_{x''}) \rangle + (\overline{H}_x - \overline{H}_{x'})(\overline{H}_x - \overline{H}_{x''}) \\
&=: \langle \epsilon_x^2 \rangle + R_x + \overline{R}_x
\end{aligned}
\tag{41}
$$

The angle brackets refer to averages over different realisations of the background state and data source errors. As one can see, the estimation of $\langle \epsilon_x^2 \rangle$ is affected by an error, which has two components. The term $R_x$ is related to correlations of wave height

differences in the background statistics. In situations where all three data sources are in a region with a spatial wave height gradient, typically observed in coastal areas, this term will not vanish, at least as long as the data sources are located along the

gradient. The term $\overline{R}_x$ is related to the differences in mean wave heights at the three locations. This term can be expected to contribute to the estimation error in coastal areas as well.

We are now estimating these error contributions for the background statistics derived in Section 2.3. Lets assume that the wave height along a straight line between the stations "HEL" and "ELB" can be approximated reasonably well with a linear function, i.e.,

$$H_x = \frac{x(H_{Hel} - H_{Elb}) + dH_{Elb}}{d} \;, \tag{42}$$

where $x$ denotes the distance of some point $X$ from the Elbe station in the direction of Helgoland, $H_{Hel}, H_{Elb}$ are the wave-heights at the Helgoland and Elbe stations, and $d = 24$ km is the distance between the two stations. Defining $\hat{H}_{Hel}$ and $\hat{H}_{Elb}$ analogues to eq. 40, we get for the wave height covariance of two points $x, x'$

$$
\begin{aligned}
\langle \hat{H}_x \hat{H}_{x'} \rangle &= \frac{1}{d^2} \langle \left( x(\hat{H}_{Hel} - \hat{H}_{Elb}) + d\hat{H}_{Elb} \right)\left( x'(\hat{H}_{Hel} - \hat{H}_{Elb}) + d\hat{H}_{Elb} \right) \rangle \\
&= xx' \langle \frac{(\hat{H}_{Hel} - \hat{H}_{Elb})^2}{d^2} \rangle + (x + x')\langle \hat{H}_{Elb} \frac{(\hat{H}_{Hel} - \hat{H}_{Elb})}{d} \rangle + \langle \hat{H}_{Elb}^2 \rangle \\
&=: xx'\alpha_1 + (x + x')\alpha_2 + \alpha_3 \;,
\end{aligned}
\tag{43}
$$

where $\alpha_1 = 0.000147$, $\alpha_2 = 0.003174$ m, and $\alpha_3 = 1.7529 \, \text{m}^2$ for the considered case. With this information eq. 41 can be evaluated. For the spatial distribution of the three data sources we assume that $x = 10$ km, $x' = 5$ km, and $x'' = 15$ km, i.e., all three data sources are within 10 km distance. The resulting errors for the *triple collocation* method then follow as (see eq. 41):

$$R_x^{10km} + \overline{R}_x^{10km} = -0.003675 \text{m}^2 - 0.00035 \text{m}^2 = -0.0040 \text{m}^2 \tag{44}$$

If an instrument at location $x$ is considered, which has a "truth" observation error standard deviation of $\sqrt{\langle \epsilon_x^2 \rangle} = 0.1$ m, these error terms would lead to an estimation error by the *triple collocation* method of about 40% in terms of variance. If the collocation distance is increased to 20 km, and one has data sources at $x = 10$ km, $x' = 0$ km, and $x'' = 20$ km, the following error is obtained:

$$R_x^{20km} + \overline{R}_x^{20km} = -0.01473 \text{m}^2 - 0.00138 \text{m}^2 = -0.0161 \text{m}^2 \tag{45}$$

In this case the collocation error grows to 160% with respect to the "truth" observation error. As explained in Section 2.1, the multi collocation method proposed in this study is designed to take spatial gradients as discussed above into account, however at the cost of requiring a larger number of data sources. This will be illustrated in Section 2.5 using the same background statistics.

The second issue to be discussed in this section, is the role of the spatial resolution of the different models and observations. The main point to consider here, is that subresolution variations of waveheight become part of the estimated data set error, if the triple or multi collocation methods are applied. This has two main consequences:

– The estimated data source errors are influenced by the background statistics.

– For two data sources with a common unresolved band of spatial scales, the data source errors are correlated (Vogelzang and Stoffelen, 2012).

In general, the expected unresolved subresolution variance of wave height is given by

$$H_{sub}^2 = \frac{1}{A} \int\limits_A \langle (H_{x'} - \overline{H}_x^{data})^2 \rangle dx' \ , \tag{46}$$

where $A$ is the resolution cell of the assumed data source and the respective data source wave height $\overline{H}_x^{data}$ is computed as

$$\overline{H}_x^{data} = \frac{1}{A} \int\limits_A H_{x'} dx' \ , \tag{47}$$

where $H_{x'}$ is the "truth" wave height at location $x'$ within the resolution cell. We now evaluate these terms, again using the background statistics presented in Section 2.3. For simplicity, we assume that the resolution cell is one-dimensional and spans from the Elbe station ($x$=0) to some point $x = a$ in the direction of Helgoland. The data set then corresponds to averages of the form

$$\overline{H}_{a/2}^{data} = \frac{\frac{a}{2}(H_{Hel} - H_{Elb}) + dH_{Elb}}{d} \ . \tag{48}$$

The mean unresolved variance within one resolution cell can then be written as

$$
\begin{aligned}
H_{sub}^2 &= \frac{1}{a} \int\limits_0^a \langle \left( \frac{x'(H_{Hel} - H_{Elb}) + dH_{Elb}}{d} - \overline{H}_{a/2}^{data} \right)^2 \rangle dx' \\
&= \frac{1}{a} \int\limits_0^a \langle \left( \frac{(x' - a/2)(H_{Hel} - H_{Elb})}{d} \right)^2 \rangle dx' \\
&= \frac{a^2}{12} \left( \frac{\langle (\hat{H}_{Hel} - \hat{H}_{Elb})^2 \rangle}{d^2} + \frac{(\overline{H}_{Hel} - \overline{H}_{Elb})^2}{d^2} \right) \ .
\end{aligned} \tag{49}
$$

One can see, that the mean sub-resolution variance is depending on the mean gradient, as well as the variance of the gradient within the resolution cell. Using the background statistic values in Section 2.3, the variance $H_{sub}^2$ was computed for different values of the cell size $a$. For $a = 5$ km one gets $H_{sub}^2 = 0.0003\,\mathrm{m}^2$, for $a = 10$ km the result is $H_{sub}^2 = 0.0012\,\mathrm{m}^2$, and $a = 20$ km gives $H_{sub}^2 = 0.005\,\mathrm{m}^2$. Lets imagine an observation instrument located at $x = a/2$ with a measurement error standard deviation of 0.1 m. This value is supposed to only describe the instrumental errors, i.e., the errors that one always has, even if the wave height within the resolution cell is constant. Furthermore, assume that the observations represent averages over waveheights in the resolution cell of size $a$ as described by eq. 48. For $a = 10\,\mathrm{km}$ this averaging process adds about 10% to the data set error variance, and for $a = 20\,\mathrm{km}$ this increases to 50%.

The above analysis has shown, that both the collocation distance and the spatial data set resolutions are important factors for the quantification and interpretation of the respective data set errors. The separation of instrumental errors and subresolution related errors is a challenge, because it requires knowledge about the "truth" background wave statistics on a subresolution scale. In general, such information can only be obtained, if one of the data sources has a significantly higher spatial resolution than the other data sources.

## 2.5   Monte Carlo simulation for 1D case

As an example, we consider the case where we have data sources, which are approximately located along a straight line. This corresponds to the scenario depicted in Fig. 3b. We approximate the truth state by a linear model with two parameters. From eq. 9 it follows, that we need at least 5 data sources to estimate the errors. Lets assume, we have two buoys, a satellite altimeter with two measurements close to the buoys, and a numerical model estimate in the middle between the two buoys. Using the wave heights at the buoy positions as the state parameters $\mathbf{t}$ one gets

$$A = \begin{pmatrix} 1 & 0 \\ 0 & 1 \\ 1/7 & 6/7 \\ 6/7 & 1/7 \\ 1/2 & 1/2 \end{pmatrix} \tag{50}$$

for the matrix $A$, which relates the truth vector to the observations (see eq. 4). Here, we have assumed a geometry as depicted in Fig. 3b. The first and second row of $A$ refer to the two buoy measurements, which are assumed to be without systematic errors. The third and fourth row correspond to the two altimeter measurements near the "ELB" buoys and the "HEL" buoy, which are assumed to be affected by calibration errors with scaling parameters of 1.2 and 1.3. The last row represents the wave height estimate provided by the wave model in the middle between the two buoys. The model is assumed to have a calibration error with a factor 0.9.

The Monte Carlo experiments were then performed as follows:

- 120 observation vectors $y$ were created using a random simulator with prescribed variances and covariances for the background statistics and the observation errors.

- The observation errors and their uncertainty was estimated using the approach described in Section 2.1.

- These experiments were repeated 1000 times to obtain statistically robust results.

The parameters used for the simulations, as well as the obtained results are summarised in Table 3. The first three columns refer to the assumed observation error statistics for the buoys, the altimeter, and the numerical model. One can see, that a covariance of 0.056 m$^2$ was used for the two satellite measurements, which corresponds to an error correlation of 0.5. The last three columns refer to the estimation errors, which were obtained in two different ways:

- The uncertainties are estimated directly by computing the variance of the estimated observation errors over all experiments. This is called "averaged experiments" approach (AVEXP) in the following.

- The uncertainties were estimated for each experiment from the input data covariance matrices as explained in Section 2.1. These estimates were than averaged over all experiments. This is called "covariance matrix" approach (COMAT) in the following.

For the obtained data source errors averaged over all experiments, the numbers agree with the assumed errors within 3 decimals, which illustrates the validity of eq. 12. The same is also true for the estimated uncertainties for the variances and covariances estimated from eq. 20. The last three columns in Table 3 show that the covariance matrix method and the numbers from the averaged experiments are in very good agreement. The last column contains the respective comparison for the covariance of the altimeter measurement errors, where the two approaches also give very consistent results. Overall, these results confirm that the estimation of uncertainties in the estimated stochastic errors by eq. 20 is a reasonable approach.

In a second step the same excercise was done for the estimation of the systematic errors. The first column of Table 4 shows the assumed calibration errors, i.e., scaling parameters used in the generation of the synthetic observations. In this case the estimated calibration factors averaged over all experiments shown in the second column agree with the theoretical values within two decimals, which seems reasonable. The values for the estimation errors obtained with the COMAT approach (fourth column) and the AVEXP approach (third column) are also in good agreement, considering that several approximations (e.g., eq. 38) were used.

## 3   Description of data sets

In this section the observation and numerical model data used for the multi collocation analysis are introduced. The data sets are from the period April 2016 to August 2017.

### 3.1   Satellite altimeter data

The spaceborne data used here were taken by the European satellite Sentinel-3A launched in February 2016. The satellite flies on a sunsynchronous orbit with an exact repeat cycle of 27 days. The spatial accuracy of the revisit is $\pm 1\,\mathrm{km}$ in longitudinal direction. Among other instruments, the platform hosts a radar altimeter (SRAL) operating at Ku- and C-band (Le Roy et al., 2007). The main frequency used for range measurements is in the Ku-band (13.575 GHz), while the C-band frequency (5.41 GHz) is used for ionospheric correction. The basic footprint of the altimeter antenna is a disc with approximately 20 km diameter. However, the effective area actually influencing the measurements is more narrowly centred around the nadir point with a diameter of about

$$A = \frac{\pi R_0 (c\tau + 2H_s)}{1 + R_0/R_e} \quad . \tag{51}$$

Here, $R_0$ = 814 km is the altitude of the satellite, $R_e$ is the radius of the earth, $c$ is the speed of light, $\tau$ is the pulse duration, and $H_s$ is significant wave height (Chelton et al., 1989). For the typical pulse durations in the order of 3 ns, the effective footprint varies between 1km and 10km with larger footprints at high sea states. In particular in coastal areas, the altimeter data processing is quite involved (Chelton et al., 2001), and a number of instrument and processing parameters can have a strong impact on the characteristics of the wave height estimates.

In this study SENTINEL-3a data with 1 Hz sampling are analysed, which corresponds to measurements taken every 7 km along the track. The analysed data were acquired in the so called reduced SAR (RDSAR) mode, which is provides data

comparable to measurements from traditional satellite altimeter. A comparison of different SENTINEL-3a altimeter modes can be found in Wiese et al. (2018).

Fig. 2a shows the distribution of SENTINEL-3a tracks over the North Sea. 'Ascending' passes are from South-south-east to North-north-west, whereas 'descending' passes are from North-north-east to South-south-west

## 3.2 In-situ Measurements

In this study insitu wave height measurements distributed over the Global Telecommunications System (GTS) were used, which are archived at the European Centre for Medium-Range Weather Forecasts (ECMWF) (Bidlot and Holt, 2006). Additional wave observation data were gathered by ECMWF as part of the JCOMM Forecast Verification project (Bidlot et al., 2002). These measurements have a quite inhomogeneous geographical distribution as shown in Fig. 2. As one can see, the focus of the observations is on coastal areas and regions with intense offshore activities, like the northern part of the North Sea. Some of the insitu stations shown in Fig. 1 and Fig. 2b, which are referenced in the subsequent analysis, are labeled by either 5 digit numbers (e.g., "62168") or three character strings (e.g., "ELB"). Due to the lack of respective metadata, it was not possible to distinguish between different types of instruments, e.g., waverider buoys or platform mounted devices. One exception is the station "62170" near the east English Channel entrance, which is identical to the light ship "F3" mentioned in Anderson et al. (2016). In addition to the GTS data, insitu wave measurements taken in the German Bight were obtained from the Bundesamt für Seeschifffahrt und Hydrographie (BSH). The GTS data have a temporal sampling of 1 hour, while the BSH buoys provide observations every 30 minutes. The insitu observations represent raw values and were checked for unrealistic wave heights. Looking at all the insitu stations for the analysed period in summary, the provided significant wave heights were in the range between 0.1 m to 7.8 m. These are realistic values for the North Sea (Semedo et al., 2015) and hence all observations were used in the analysis.

## 3.3 Wave Model WAM and meteorological input data used

For this study, data generated with the spectral wave model WAM were used (Komen et al., 1996). The model version Cycle4.6.2 considered here includes depth refraction and wave breaking and is therefore suitable for coastal applications (Staneva et al., 2017). Spatial variations in bathymetry are taken into account, however temporal variations of water depth due to tides are not included in the simulations. The 2d-wave spectra are calculated on a polar grid with 30 directional $15°$ sectors and 30 logarithmically spaced frequencies ranging from 0.042 to 0.66Hz. A spherical grid is used for the space dimensions with $\sim 0.06°$ resolution in zonal and $\sim 0.03°$ resolution in meridional direction. The required forcing at the open boundaries of the North Sea model domain are derived from a coarser model simulation for the whole North Atlantic. Model output with 1 hour time steps was available for the analysis. ERA-5 data are used as meteorological forcing for the North Sea model runs (Hersbach and Dee, 2016). This data set is a global re-analysis product from ECMWF with a spatial resolution of of $31\,\mathrm{km}$. The model results are interpolated to a $0.25°$ grid, and the time step is one hour in the final product. A detailed comparison of different model setups with satellite altimeter data can be found in Wiese et al. (2018).

Compared to previous studies (Janssen et al., 2007; Caires and Sterl, 2003), the spatial resolutions of the three analysed data sources are in quite close agreement. The effective resolutions of the altimeter and the insitu instruments both depend on the actual sea state. For the altimeter typical footprint sizes are between 1 km and 10 km as explained in Section 3.1. For the insitu data, the translation of the typical 20 min averages to spatial averages is determined by the group velocity. For example, the energy will propagate with about 15 km/hour, if the dominant wave length is 50 m long and the water is deep (>50 m). A 20 min temporal average would therefore correspond to a 5 km spatial average in this case, which is in good correspondence to the spatial model resolution of about 3.5 km. We have therefore used the original data for the analysis and not generated super-observations by averaging, as done in Janssen et al. (2007) and Caires and Sterl (2003), who used wave model data with significantly coarser resolution.

## 4 Triple collocations for the entire North Sea

In this section the triple co-location method, as a special case of the multi collocation approach, is applied to the SENTINEL-3a altimeter wave height measurements introduced in Section 3.1 to assess the respective systematic and stochastic errors. The novelty lies in the analysis of a new satellite data set and the provision of error bars for the estimated stochastics and systematic errors.

Traditionally, validations of new data sets are performed by comparing to data from established in-situ measurements, which are regarded as a reference. Here, the following assumptions are made

- Sentinel-3a and the WAM model maybe affected by calibration problems represented by the calibrations factors $\lambda_{S3}, \lambda_{WAM}$.

- Sentinel-3a and the WAM model maybe affected by biases $b_{WAM}, b_{S3}$.

- Buoys are regarded as reference systems, i.e., they are assumed as bias free and without calibration errors

Each of the SENTINEL-3a tracks shown in Fig. 2a is passed by the satellite about once a month. Fig. 2b shows the respective number of co-locations found, if a maximum distance of 10 km is accepted. The co-location involves some necessary interpolation steps, which were performed as follows (Janssen et al., 2007):

- The model is interpolated to the buoy using linear interpolation.

- The model is interpolated to the closest altimeter point using linear interpolation.

- Both the buoy and the model are interpolated to the satellite overflight time.

- The model value used for the location is taken as the average of the buoy and the satellite interpolation (see Janssen et al. (2007)).

The *triple collocation* technique was applied to each insitu platform, for which altimeter data within the acceptable collocation could be found. The direct method as described in Section 2.2 was used for this analysis.

As an example, Fig. 5 shows the obtained results for the Elbe buoy "ELB" located at 54.0°N 8.1°E. The location of this buoy can also be found in Fig. 1. The three scatter plots show the used data sets in different combinations (buoy versus WAM (a), buoy versus SENTINEL-3a (b), and WAM versus SENTINEL-3a (c)). The three data sets were corrected according to the slope and bias parameters estimated in the collocation procedure. The slope parameters for both the model and SENTINEL-3a were

5 found to be below 1, and there exists a larger positive bias for the altimeter. The red triangles correspond to ascending satellite passes and green triangles indicate descending satellite heading. A connection between the satellite flight direction and errors is not clearly visible. This is an important result, because the altimeter data processing is known to be more challenging for passes going from land to sea. It is evident, that the best agreement is between the buoy and the model. The smallest stochastic error is found for the buoy with 0.04 m standard deviation. For this location, the collocation procedure gives the largest stochastic

error of 0.25 m for the altimeter data.

Fig. 6 shows the estimated biases (left column) for the SENTINEL-3a altimeter (top) and the wave model (bottom). One can see, that the altimeter seems to be either bias free or slightly biased high for most of the cases (Fig. 6a). Averaging over all buoys, gives a bias estimate of

$$\langle b_{S3} \rangle_{buoys} \approx 0.07 \pm 0.31 \text{ m} . \tag{52}$$

Concerning the spatial distribution of observation errors, it is hard to draw conclusion. It is however evident, that the few cases with low bias are far offshore (Fig. 6a) For the wave model there a more cases, where a small low bias is found (Fig. 6d). Again averaging over all buoys gives

$$\langle b_{WAM} \rangle_{buoys} \approx -0.03 \pm 0.26 \text{ m} . \tag{53}$$

The spatial distribution shows a weak clustering of low bias cases in the northern part of the North Sea. It is interesting to

20 note, that for the location of the lightvessel "62170" near the east entrance of the English Channel (see Fig. 1) the satellite and the model show a positive bias of about 0.3 m and 0.2 m respectively. According to Anderson et al. (2016), one can expect a systematic low bias for wave height measurements from lightvessels of about 0.3 to 0.4 m. It is thus possible, that the estimated high bias for satellite and model is in this case an artefact caused by the violated assumption of bias free insitu observations.

The scaling parameter for the satellite altimeter shown in Fig. 6b indicates values above 1 for most of the cases. In fact,

averaging over all buoys gives

$$\langle \lambda_{S3} \rangle_{buoys} \approx 1.11 \pm 0.27 . \tag{54}$$

The respective scaling parameter estimation errors derived using the approach described in Section 2.2 are shown in Fig. 6c. It is evident, that quite a few of the cases with exceptionally high scaling values (around 1.2) are affected by large estimation errors. This is a good illustration of the added value provided by the error estimation procedure presented in this study. The

30 corresponding scaling parameters for the WAM model shown in Fig. 6e show values, which are closer to unity for the most part. The respective mean value is

$$\langle \lambda_{WAM} \rangle_{buoys} \approx 1.02 \pm 0.20 \tag{55}$$

with higher values (around 1.1) found in the English Channel. Most of the other cases closer to the coast have slope values slightly below unity. Most of the cases with large estimation errors for the scaling factor (Fig. 6f) are found close to the coast.

Results for the stochastic errors are summarised in Fig. 7. The columns refer to SENTINEL-3a (left column), the WAM model (centre column), and the buoys (right column). The top row shows the estimated stochastic error standard deviation and the bottom row the respective relative estimation errors $\nu$, defined as

$$\nu = 100\% \, \frac{\text{stdv}(\langle \epsilon^2 \rangle)}{\langle \epsilon^2 \rangle} \, , \tag{56}$$

where $\langle \epsilon^2 \rangle$ is the error variance, and stdv($\langle \epsilon^2 \rangle$) is the standard deviation of the respective estimator, derived using the approach described in Section 2.1. One can see, that overall, the smallest stochastic errors are found for the buoys, as expected (Fig. 7c). In fact, one gets

$$\langle \langle \epsilon^2_{Buoy} \rangle^{1/2} \rangle_{buoys} \approx 0.12 \pm 0.11 \text{ m} \tag{57}$$

averaging over all buoys. There are two buoys, which stand out with errors above 0.25 m, in the northern part of the North Sea. In this case the estimation errors are not exceptionally high and possible reasons for these relatively high error levels should be further investigated. In general, one can see that the estimations errors are quite large, exceeding in most cases 20% (Fig. 7f).

The stochastic errors of the WAM model (Fig. 7b) and the altimeter (Fig. 7a) are quite similar in their average values

$$\langle \langle \epsilon^2_{WAM} \rangle^{1/2} \rangle_{buoys} \quad \approx \quad 0.17 \pm 0.07 \text{ m} \tag{58}$$

$$\langle \langle \epsilon^2_{S3} \rangle^{1/2} \rangle_{buoys} \quad \approx \quad 0.18 \pm 0.14 \text{ m} \tag{59}$$

It is interesting to see, that the two buoys mentioned above also stand out with respect to the corresponding model errors. Theoretically, this could be due to a correlation between the background statistics and both the model and buoy errors. However, because this is observed in a quite homogeneous offshore area, with neighboring buoys not showing the same effect, this explanation is not very likely. It is more likely, that the basic assumptions about zero bias or absent calibration errors are violated for these buoys.

The finding that, on average, the insitu stations have the smallest stochastic errors is at first sight in disagreement with results presented in Janssen et al. (2007). One has to take into account however, that there are a number of significant differences in the analysis. First of all, a global wave model with 55 km resolution was used in the former study, whereas the computational model grid used in our analysis has a resolution more than 15 times higher. It is unlikely however, that the coarser model resolution is the only factor, because Caires and Sterl (2003) also concluded, that the insitu stations have the smallest stochastic errors using wave model output with even coarser resolution (1.5°) than used by Janssen et al. (2007). Both studies introduced altimeter super-observations (averages over subsequent measurements) to make the altimeter observations more consistent with the model estimates. In the present study this was not considered necessary, because the altimeter and model resolutions are in much closer agreement. The second major difference with respect to previous studies is the geographic locations and the type of altimeter data considered in the analysis. Janssen et al. (2007) investigated global ERS-2 and ENVISAT altimeter data sets, while Caires and Sterl (2003) concentrated on TOPEX and ERS-1 altimeter data acquired over the Pacific and the US

east coast. This means that there are certainly differences both with regard to the background wave statistics and the satellite and insitu observation errors. A third important difference between the studies is the applied collocation criteria. Janssen et al. (2007) required the model, insitu and satellite estimates to be within 200 km distance and Caires and Sterl (2003) used a smaller collocation distance of 0.75°. The allowed distance of 10 km used in the present study is still significantly smaller than that, and the collocation errors are therefore also likely to be smaller. For the above reasons one cannot conclude that the present study contradicts the results in Janssen et al. (2007). The conclusion is rather, that a common set of reference insitu data and collocation criteria are desirable to make different studies more comparable.

It is evident that the observed heterogeneity of insitu measurements is a big complicating factor in the analysis. Wave model computations and satellite altimeter observations have reached a level of accuracy, where further improvements require a very careful selection and treatment of validation data sets. This in particular requires more knowledge about the type of insitu instruments and applied data processing techniques (e.g., averaging intervals). This could also be an argument for investments into dedicated validation instruments with more transparent and better documented error characteristics and quality control. The deployment of such instruments should take into account both research aspects and requirements for operational use.

## 5 Multi collocations

In this section different examples are presented, where more than 3 observations are combined, i.e., this is beyond the standard *triple collocation* approach. The two example discussed in the following are typical situations encountered, when analysing insitu data, model data and satellite measurements in combination.

### 5.1 1D example

The geometry of the first example is depicted in Fig. 8a. Here, an ascending SENTINEL-3a track passes between the two insitu stations "62150" and "62289". The station on the easterly side is within 10 km distance of the track and was therefore used in the *triple collocation* study presented in Section 4. Station "62150" did not match the criteria and was disregarded for the analysis. Both stations can be found in Fig. 1 and Fig. 2b, where they are indicated by triangle symbols.

The idea to relate both insitu measurements to the altimeter track, is to use a linear interpolation of the "truth" wave height between the two stations, which makes the use of the instrument with the larger distance more acceptable in the collocation procedure. In principle, this corresponds to the 1d case depicted in Fig. 3b with the role of altimeter and model interchanged. In the present situation there is one altimeter measurement between the two reference instruments and for simplicity the numerical model wave height estimate is taken at the location of the buoys. Because of the small number of available samples, we have also used altimeter measurements, which are slightly above and below the connecting line (red dots in Fig. 8a). This resulted in $n_s$=14 common data samples that could be used for the statistics.

Using this geometry, allows estimation of the errors of all data sources, as well as the error correlations between the model wave heights (see table 1). The calibration factors and their respective standard deviations were estimated with the direct and

iterative method and are as follows:

$$
\begin{aligned}
\lambda^{WAM}_{62150} &= 0.662 \pm 0.147 \ (0.788 \pm 0.161) \\
\lambda^{WAM}_{62289} &= 0.779 \pm 0.113 \ (0.778 \pm 0.100) \\
\lambda_{S3} &= 1.023 \pm 0.246 \ (1.023 \pm 0.360)
\end{aligned}
\tag{60}
$$

The values in brackets were obtained with the iterative method. Significant differences are only found for the first scaling parameter. However, both methods provide consistent results, if the error bars are taken into account. It is interesting to note, that the scaling value for SENTINEL-3a is closer to unity than the smaller value of about 0.8 found by the *triple collocation* method (Fig. 6b). This value was exceptional among the other buoys, for which numbers above 1 were found for the most part. This could be an indication for a problem with station "62289", which also stands out in the stochastic errors shown in Fig. 7c. The numbers obtained for the stochastic errors are as follows:

$$
\begin{aligned}
\mathrm{var}(\epsilon^{Buoy}_{62150}) &= -0.0890 \pm 0.0914 \,\mathrm{m}^2 \ (-0.0889 \pm 0.0781 \,\mathrm{m}^2) \\
\mathrm{var}(\epsilon^{Buoy}_{62289}) &= -0.0072 \pm 0.0234 \,\mathrm{m}^2 \ (-0.0072 \pm 0.0235 \,\mathrm{m}^2) \\
\mathrm{var}(\epsilon^{WAM}_{62150}) &= 0.0749 \pm 0.0467 \,\mathrm{m}^2 \ (0.0913 \pm 0.0557 \,\mathrm{m}^2) \\
\mathrm{var}(\epsilon^{WAM}_{62289}) &= 0.0234 \pm 0.0167 \,\mathrm{m}^2 \ (0.0234 \pm 0.0167 \,\mathrm{m}^2) \\
\mathrm{covar}(\epsilon^{WAM}_{62150}, \epsilon^{WAM}_{62289}) &= 0.0242 \pm 0.0095 \,\mathrm{m}^2 \ (0.0241 \pm 0.0095 \,\mathrm{m}^2) \\
\mathrm{var}(\epsilon_{S3}) &= 0.1372 \pm 0.0555 \,\mathrm{m}^2 \ (0.1372 \pm 0.0550 \,\mathrm{m}^2)
\end{aligned}
\tag{61}
$$

It can be seen, that the estimates for the buoys are slightly negative, which is not meaningful for a variance. This can in fact happen for small sample sizes, since the estimators do not guarantee positive values. In this case it is helpful to look at the respective error bars, which are given as standard deviations. For a Gaussian distributed variable the interval given by $\pm$ stdv gives the 68% confidence interval, i.e., more than 30% of the cases are outside of this value range. This means, that the estimated values for buoys are consistent with small positive error variances. The largest value is found for the SENTINEL-3a altimeter with a relatively small error bar. This is consistent with the finding already made with the *triple collocation* method (see Fig. 7a). For the WAM model at the location of the "62289" station, the triple collocation method gave a similarly high value, but with almost 100 % error margin. The estimate obtained with the multi collocation is significantly lower, but again with a large relative estimation error of about 100 %. Because of the smaller mean value, the latter estimates still point towards a smaller model error, than indicated by the *triple collocation* method.

The covariances estimated for the WAM wave height errors at the two buoy locations corresponds to a correlation value of 0.58. If we assume that the error autocorrelation function is Gaussian shaped, i.e.,

$$
ACF(\Delta x) = \exp(-\frac{\Delta x}{\lambda_C})
\tag{62}
$$

with correlation length $\lambda_C$ and spatial distance $\Delta x$, the above value results in $\lambda_C$ = 55 km.

Because of

$$
\langle(\epsilon^{WAM}_{62150} - \epsilon^{WAM}_{62289})^2\rangle = \mathrm{var}(\epsilon^{WAM}_{62289}) + \mathrm{var}(\epsilon^{WAM}_{62150}) - 2\,\mathrm{covar}(\epsilon^{WAM}_{62289}, \epsilon^{WAM}_{62150}),
\tag{63}
$$

the knowledge about the variances and covariances also allows an estimate of the uncertainties in the gradient estimates. In this case an error standard deviation of 0.31 m was obtained for the difference of the WAM model wave heights at the two buoy locations.

## 5.2 2D example

The geometry of the second example is depicted in Fig. 8b. This is an area in the northern part of the North Sea around $58°$ latitude between England and Norway. In this case an ascending SENTINEL-3a track is passing through a group of three insitu wave observation platforms, which are shown in Fig. 1 and Fig. 2b. Here, we concentrate on two locations covered by the satellite, which appear as two clusters in Fig. 8b. The "North" group of satellite observations is shown in blue and the "South" group in red. Including the numerical model estimates at those locations, the situation is then as described by the last row in Table 1. One has 7 wave height estimates in total, and a 2D plane approximation is used for the observed area. The multi collocation method then allows an estimation of the errors of all components, as well as three covariances. As the buoy measurements can be assumed as independent, only two covariances are required in this example; this is the covariance between the model errors at the two locations and the same for the altimeter measurements. With this configuration the number of available data sets was $n_s$=11.

The scaling values and their standard deviations obtained with the direct method are as follows:

$$
\begin{aligned}
\lambda_{WAM}^{South} &= 1.130 \pm 0.006 \\
\lambda_{WAM}^{North} &= 1.104 \pm 0.004 \\
\lambda_{S3}^{South} &= 1.270 \pm 0.002 \\
\lambda_{S3}^{North} &= 1.272 \pm 0.003
\end{aligned}
\tag{64}
$$

Here, the values labeled with "North" refer to the northern cluster of SENTINEL-3a measurements (blue points in Fig. 8b) and the values labeled with "South" refer to the southern group of observations (red points in Fig. 8b). These estimates seem to be quite robust, because of the small error bars and the fact that the errors in both areas are very similar. The results also confirms the overall finding of the *triple collocation* analysis which indicated a wave height overestimation by the SENTINEL-3a altimeter.

The respective values for the stochastic errors and their standard deviations with the same naming convention and obtained with the direct method are as follows:

$$
\begin{aligned}
\mathrm{var}(\epsilon_{62168}) &= 0.003 \pm 0.007 \ \mathrm{m}^2 \\
\mathrm{var}(\epsilon_{62161}) &= 0.010 \pm 0.006 \ \mathrm{m}^2 \\
\mathrm{var}(\epsilon_{62134}) &= 0.014 \pm 0.007 \ \mathrm{m}^2 \\
\mathrm{var}(\epsilon_{WAM}^{South}) &= 0.016 \pm 0.008 \ \mathrm{m}^2 \\
\mathrm{covar}(\epsilon_{WAM}^{North}, \epsilon_{WAM}^{South}) &= 0.009 \pm 0.005 \ \mathrm{m}^2 \\
\mathrm{var}(\epsilon_{WAM}^{North}) &= 0.005 \pm 0.003 \ \mathrm{m}^2 \\
\mathrm{var}(\epsilon_{S3}^{South}) &= 0.011 \pm 0.007 \ \mathrm{m}^2 \\
\mathrm{covar}(\epsilon_{S3}^{North}, \epsilon_{S3}^{South}) &= 0.005 \pm 0.005 \ \mathrm{m}^2 \\
\mathrm{var}(\epsilon_{S3}^{North}) &= 0.012 \pm 0.007 \ \mathrm{m}^2
\end{aligned}
\tag{65}
$$

Due to the significant estimation errors it is hard to tell, which data source has the smallest errors. The obtained numbers are consistent with an error standard deviation of around 0.1 m for all data sets. The error estimates for the altimeter at the two locations agree very well and are also consistent with the values found with the *triple collocation* method (Fig. 7a). The difference of the error variances for the WAM model at the two locations appear to be quite big considering the distance of about 30 km. But again, the error bars show, that there is a significant probability that the errors are actually in closer agreement. In principle, it would be possible to force the WAM error variances at the two locations to be the same, using a respective formulation of the linear system eq. 12. However, looking at the spatial variations of the bathymetry in Fig. 1, this is hard to justify.

For the correlation, a value of 0.39 was found for the altimeter and a value close to 1 for the WAM model. This corresponds to a correlation length of about 30 km for the satellite data. It makes sense that the correlation length for the WAM model is longer in this case compared to the configuration discussed in the previous section, because the analysed area is in deeper water quite far offshore, and can therefore be assumed as more homogeneous with respect to model errors.

The examples show that the multi collocation method is in fact applicable to real data source configurations. In particular, the matrix $D$ in eq. 12 is regular for the considered geometries, and estimates for error correlations can be obtained. It is also evident of course, that the limited number of samples results in significant estimation errors. According to eq. 19, the variance of the error variance estimation scales with $1/n_s$, i.e., in order to reduce the error bars given in eqs. 61 and eqs. 65 by a factor of two, the number of samples has to be increased by a factor of four.

## 6 Conclusions

The presented study provides an extension of the known *triple collocation* method, which can be useful in areas with stronger gradients, like coastal regions, where nearest neighbor approximations maybe critical. The proposed method is very flexible in

the way that various parameterisations can be used to describe the spatial variability of the measured quantities. In this study we considered only linear models, but this is not a restriction of the method, since more sophisticated functional forms (e.g., bilinear functions) can be easily integrated. Such higher order approaches are certainly desirable for coastal areas with strong spatial variations, however they require a larger number of data sources (see eq. 9).

An approach was proposed to estimate the uncertainties of estimated calibration and stochastic errors, which is also useful in the context of the standard *triple collocation* method, which is a special case of the multi collocation technique. The technique uses the covariance matrices of the input data and the number of samples as input, i.e., boot strapping is not required. These uncertainty estimates are seen as very valuable, in particular in the context of new instruments, for which only a limited data set is available for the assessment.

The proposed techniques were validated using Monte Carlo simulations with realistic background statistics. It was shown, that the obtained error estimates and their respective uncertainties are in good agreement with the expected values, although a couple of approximations had to be used in the derivation.

The method was applied to a data set of insitu wave measurements, SENTINEL-3a altimeter observations, and numerical wave model data. The number of available samples was relatively small and estimation errors had therefore to be taken into

account. The usefulness of the derived error bars for the interpretation of the data could be demonstrated. For the analysed 16 months data set presented here the estimation errors are significant, in particular if individual geographic locations are analysed. It would therefore be interesting to continue some parts of the analysis at a later stage of the SENTINEL-3a mission, when a larger data set will be available. More robust results are obtained, if the systematic and stochastic data set errors estimated for different insitu instrument locations are averaged. The results obtained for the North Sea indicate the smallest stochastic errors

for the insitu measurements, as expected. The stochastic errors of the model and the altimeter seem comparable, if averaged over all insitu locations. The analysis indicates, that on average the altimeter is overestimating wave heights by about 10% for above mean wave conditions. Two examples of multi collocations were analysed, which included a group of two and three insitu platforms respectively. In both cases a SENTINEL-3a track passed nearby, and model data were used in addition. The use of 1D and 2D parameterisations for the first and second example respectively, resulted in estimates for the spatial decorrelation

of model and altimeter errors.

The proposed method can be used for many other applications not discussed in this study. For example, it is straightforward to extend the analysis of error correlations to the time domain. The method can also be applied in situations, where different instruments do not measure exactly the same quantity, but different components of a "truth" vector, for example HF radar providing 2D current vectors and satellite SAR providing one current component (e.g., Hansen et al. (2011)).

The study is supposed to make a contribution to the optimal use of the growing number of observations, in particular in coastal areas. For applications, like data assimilation, knowledge about the errors of different data sources is essential. Analysis of observation errors is also a critical component in the design and extension of observatories used for various applications. This subject will be of growing concern, for example, in the context of the European marine core service (CMEMS), where insitu data are required to optimise forecasts for all European Seas.

**Table 1.** The number of data source error variances $n_{var}$ and covariances $n_{covar}$, that can be estimated using different dimensions of the truth parameterisation $n_t$, and number of data sources $n_o$ according to eq. 9.

|  | $n_t$ | $n_o$ | $n_{var}$ | $n_{covar}$ |
|---|---|---|---|---|
| 0d (TRIPCOL) | 1 | 3 | 3 | - |
| 1d (MULTCOL) | 2 | 5 | 5 | 1 |
| 2d (MULTCOL) | 3 | 6 | 6 | - |
|  | 3 | 7 | 7 | 3 |

**Table 2.** Mean, variance (var), covariance (covar), and correlation (corr) parameters used for the simulation of the background wave height statistics at the locations of the Helgoland and Elbe buoy in the German Bight. These numbers were derived from measurements taken during the period June 2016 - April 2017. The respective probability distributions with a log-normal approximation are shown in Fig. 4.

| Buoy | mean(log($H_s$/m)) | var(log($H_s$/m)) | covar | corr |
|---|---|---|---|---|
| Elbe | -0.109 | 0.391 | 0.354 | 0.944 |
| Helgoland | -0.014 | 0.359 |  |  |

**Table 3.** Parameters used for the Monte Carlo simulations in Section 2.5. The first two columns refer to the stochastic wave height error standard deviation (stdv) and variance (var) assumed for the considered data sources. The third column gives the assumed error cross covariance (covar) values for the two altimeter measurements and the two buoy data sets. The fourth column is the error standard deviation of the estimator for the observation error variances obtained by averaging over 1000 estimation experiments (AVEXP approach). The values in column 5 refer to the same estimation errors, but derived by application of the method described in Section 2.1 (COMAT approach). The last column gives the COMAT and AVEXP standard deviations for the covariance estimation errors.

|  | truth stdv [m] | truth var [m$^2$] | covar [m$^2$] | AVEXP stdv [m$^2$] | COMAT stdv [m$^2$] | COMAT/AVEXP stdv [m$^2$] |
|---|---|---|---|---|---|---|
| Buoy Elbe | 0.25 | 0.063 | 0 | 0.024 | 0.024 | 0 |
| Buoy Helgoland | 0.2 | 0.040 |  | 0.023 | 0.024 |  |
| Alt Elbe | 0.32 | 0.102 | 0.056 | 0.028 | 0.028 | 0.016/0.016 |
| Alt Helgoland | 0.35 | 0.122 |  | 0.025 | 0.026 |  |
| Model | 0.27 | 0.073 |  | 0.013 | 0.013 |  |

**Table 4.** Parameters used for the Monte Carlo simulations to validate the approach described in Section 2.2 for the quantification of errors in the calibration factor estimates. The MULCOL technique is applied to a 1D configuration with 5 data sources, of which three (two altimeter (Alt) observations and one model estimate) are affected by calibration errors. The first and second column give the assumed "truth" scaling parameters and the second column give the respective estimates. The last two columns represent the uncertainty estimates for the derived scaling parameters in terms of standard deviation (stdv) based on two different procedures. See text for details.

|  | truth scaling | estimated scaling | COMAT stdv | AVEXP stdv |
|---|---|---|---|---|
| Alt Elbe | 1.20 | 1.20 | 0.052 | 0.053 |
| Alt Helgoland | 1.30 | 1.30 | 0.063 | 0.063 |
| Model | 0.90 | 0.90 | 0.041 | 0.041 |

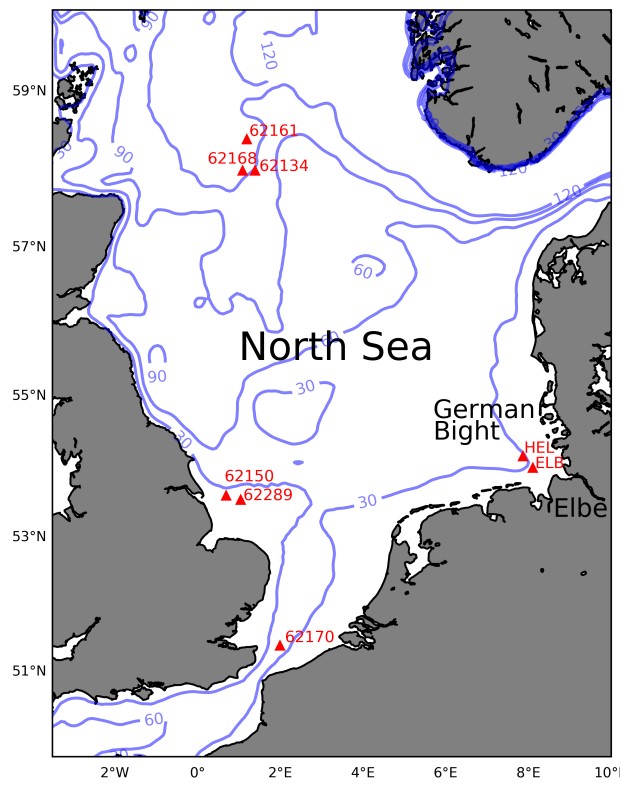

**Figure 1.** Bathymetry of the North Sea with the location of some insitu wave observation instruments considered in the study. The plot shows isobaths for 30m, 60m, 90m, and 120m water depth.

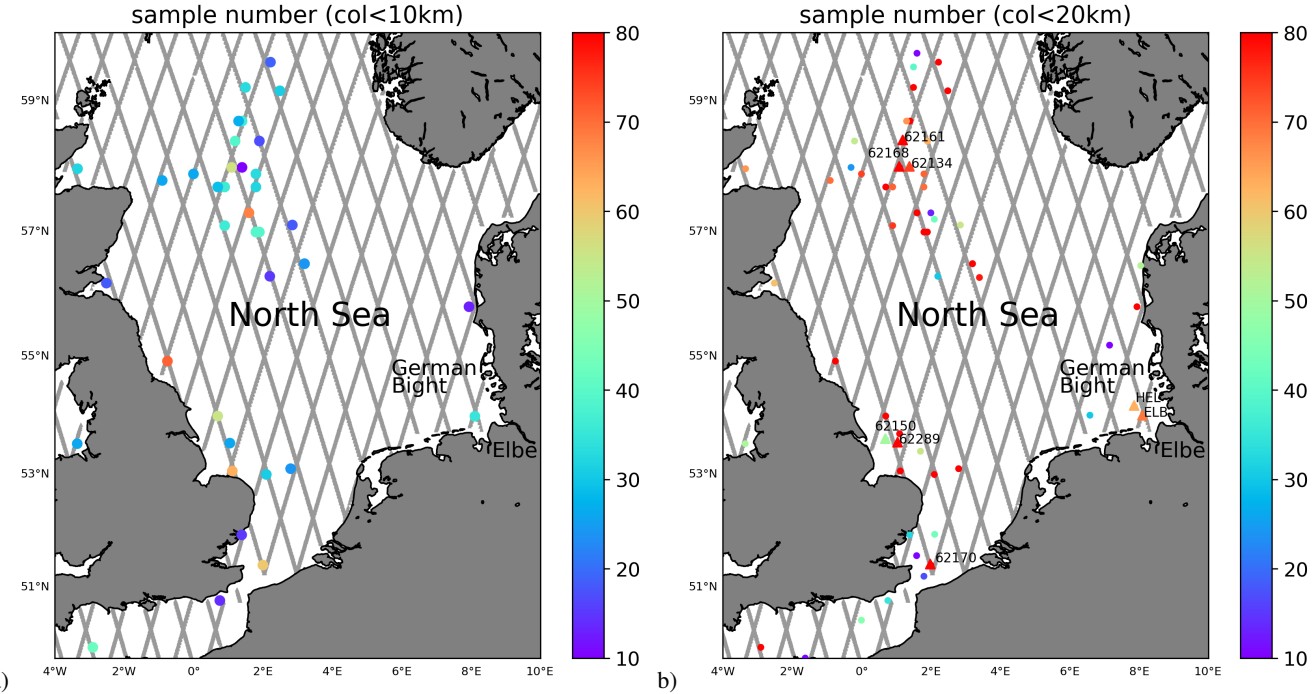

**Figure 2.** a) Map showing SENTINEL-3a altimeter tracks together with wave observation platforms with less than 10 km distance to satellite measurements. The color coding refers to the number of obtained collocated measurements in the period April 2016 to August 2017 (b) The same as a) with a collocation distance of 20 km.

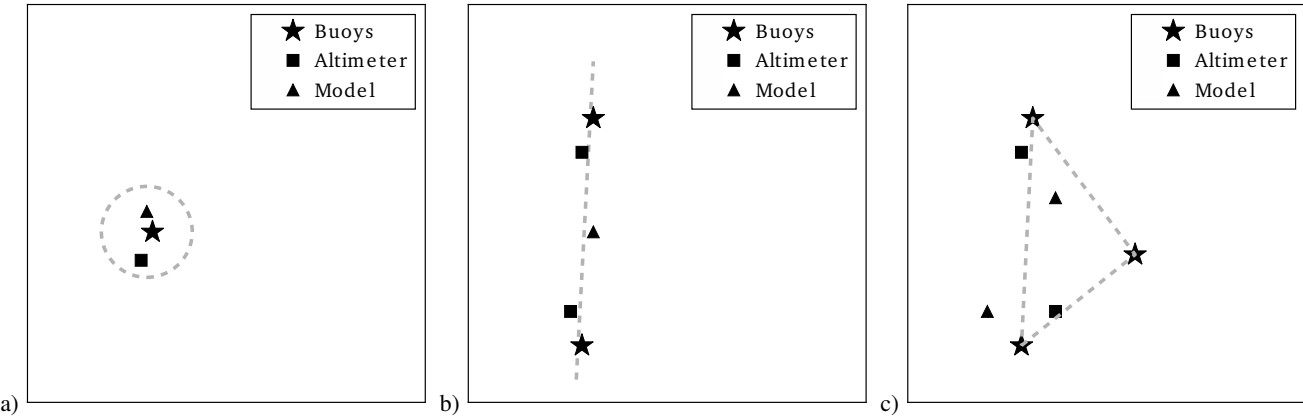

**Figure 3.** Illustration of three considered observation scenarios. (a) Data sources are assumed to provide information on the same quantity ("0d"). (b) Measurements are located along a line and a 1d linear approximation is employed for the measured quantity ("1d"). (c) The same as (b), but using a 2d interpolation method for more general spatial distributions of data sources.

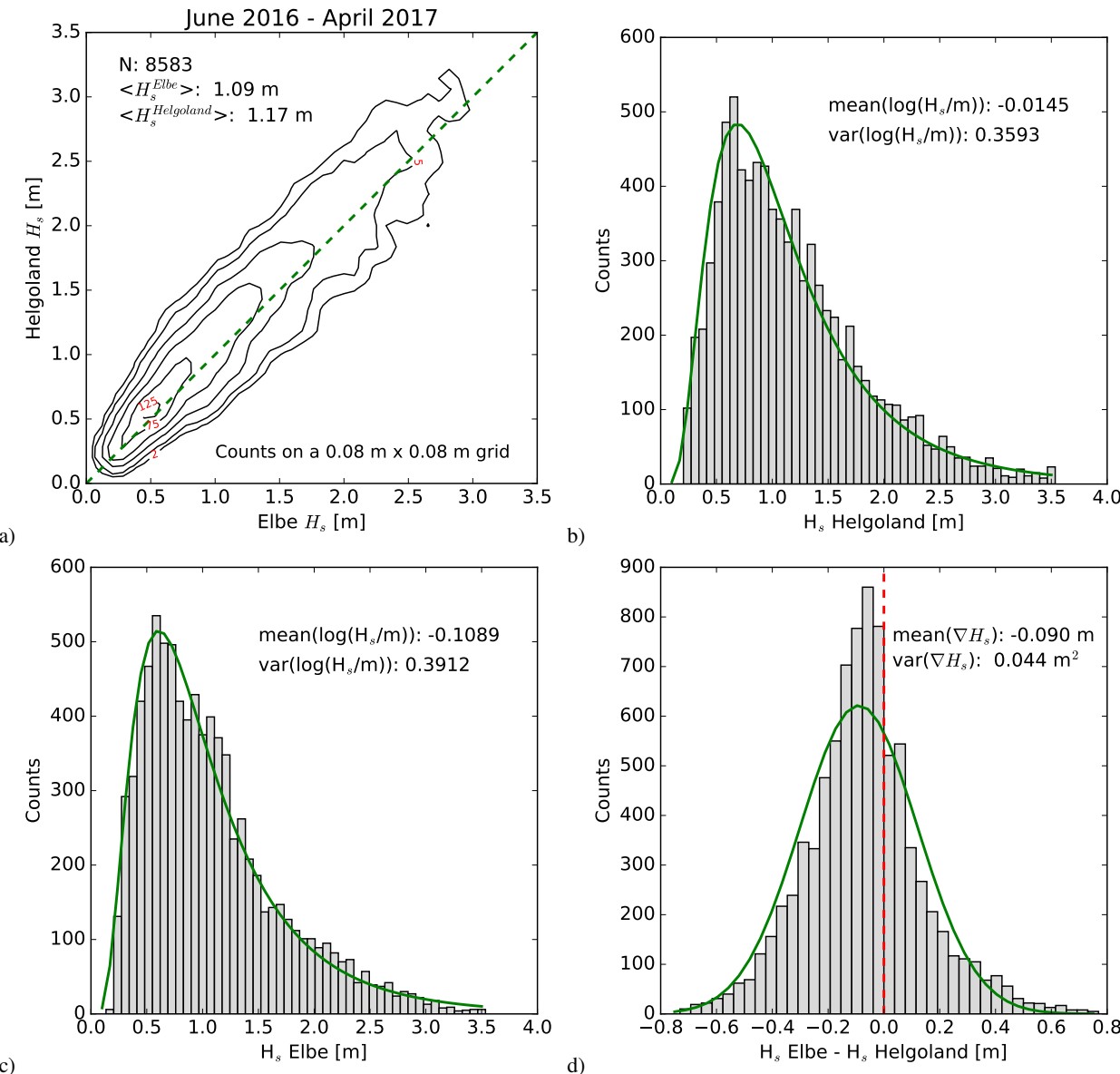

**Figure 4.** Background statistics used for the Monte Carlo Simulations. (a) 2d histogram of the joint distribution for the Elbe and Helgoland buoy derived from the period June 2016 - April 2017 with diagonal given in dashed green and the black isolines referring to probability density. (b) 1d histogram for the Helgoland buoy wave heights with log-normal pdf superimposed in green. (c) the same as b) for the Elbe buoy. (d) Histogram of the difference between the Elbe and Helgoland wave height with Gaussian pdf superimposed in green, and the dashed red line indicating the zero position.

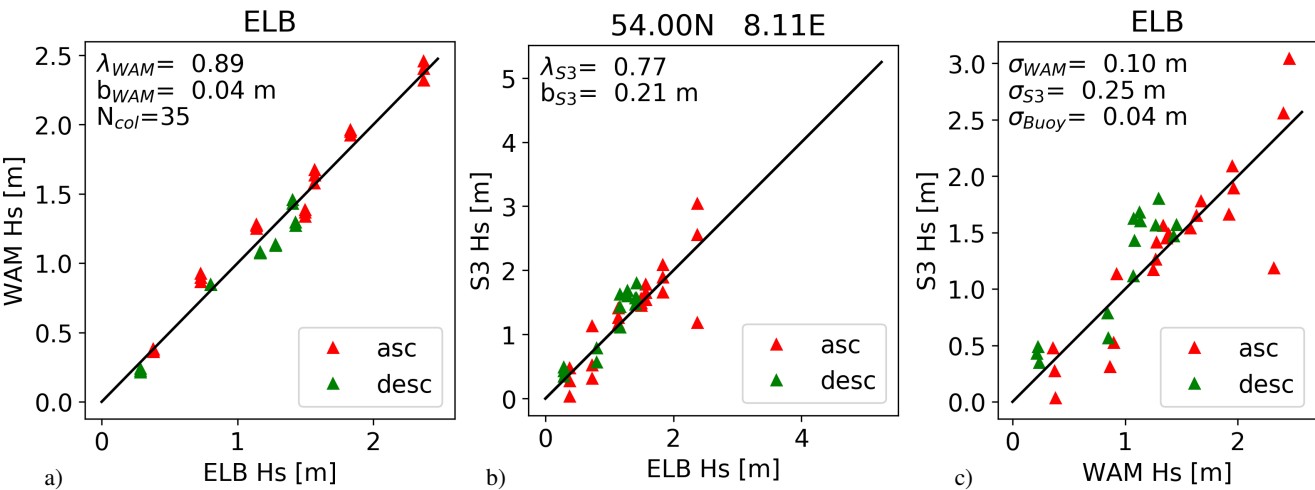

**Figure 5.** (a,b) Comparison of wave heights measured by the Elbe buoy with numerical model results (a), and SENTINEL-3a altimeter measurements (b). The model and satellite data sets were corrected according to the calibration factors $\lambda_{WAM}, \lambda_{S3}$ and bias parameters $b_{WAM}, b_{S3}$ estimated in the *triple collocation* procedure (see eq. 21). (c): Direct comparison between model and satellite data. Numbers are given for the estimated calibration factors, bias, and stochastic error standard deviations $\epsilon_{WAM}, \epsilon_{S3}, \epsilon_{Buoy}$. The red triangles refer to ascending satellite passes and the green ones to descending passes.

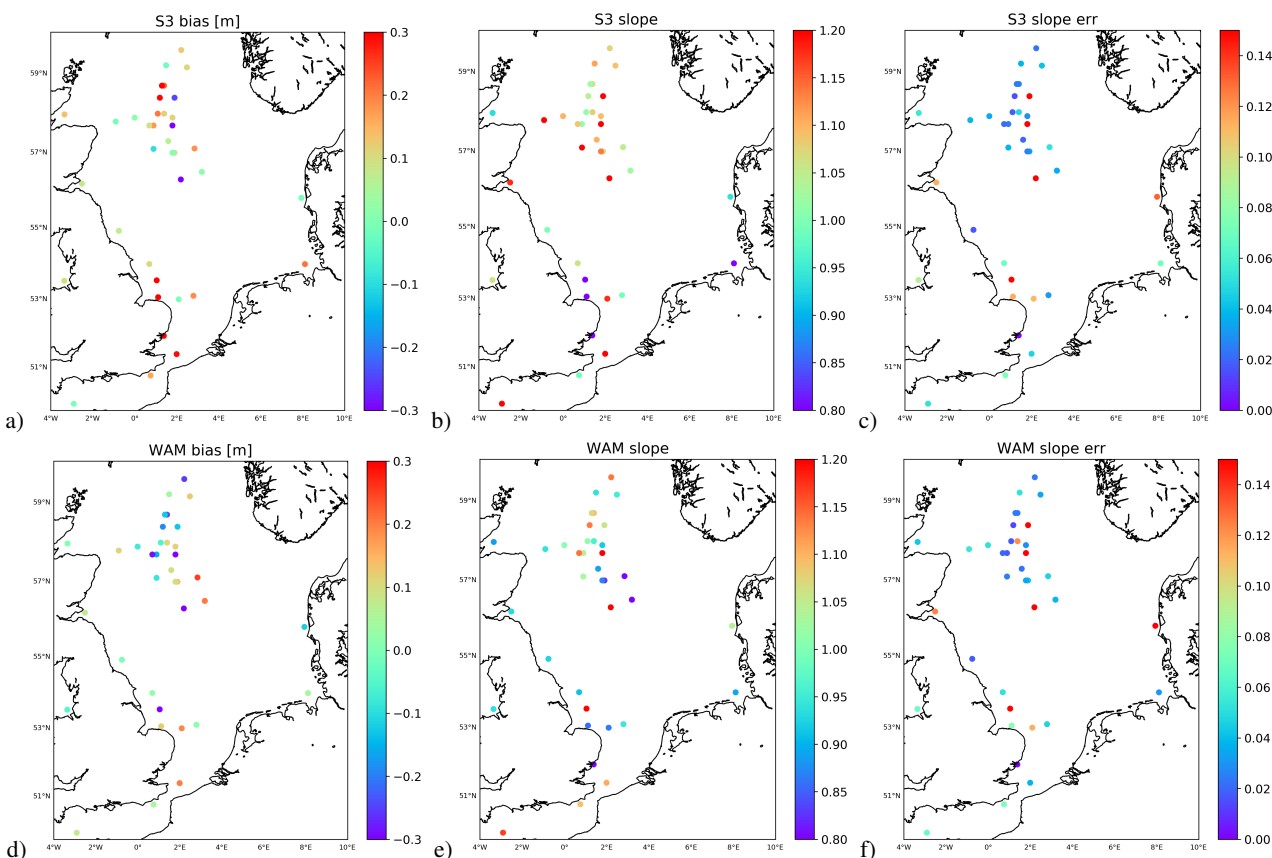

**Figure 6.** Colour coded biases (a,d) and calibration factors (b,e) derived by *triple collocation* for the SENTINEL-3a altimeter (top) and WAM model wave heights (bottom). The right column (c,f) gives the uncertainties of the slope estimations derived using the approach in Section 2.2 as standard deviation.

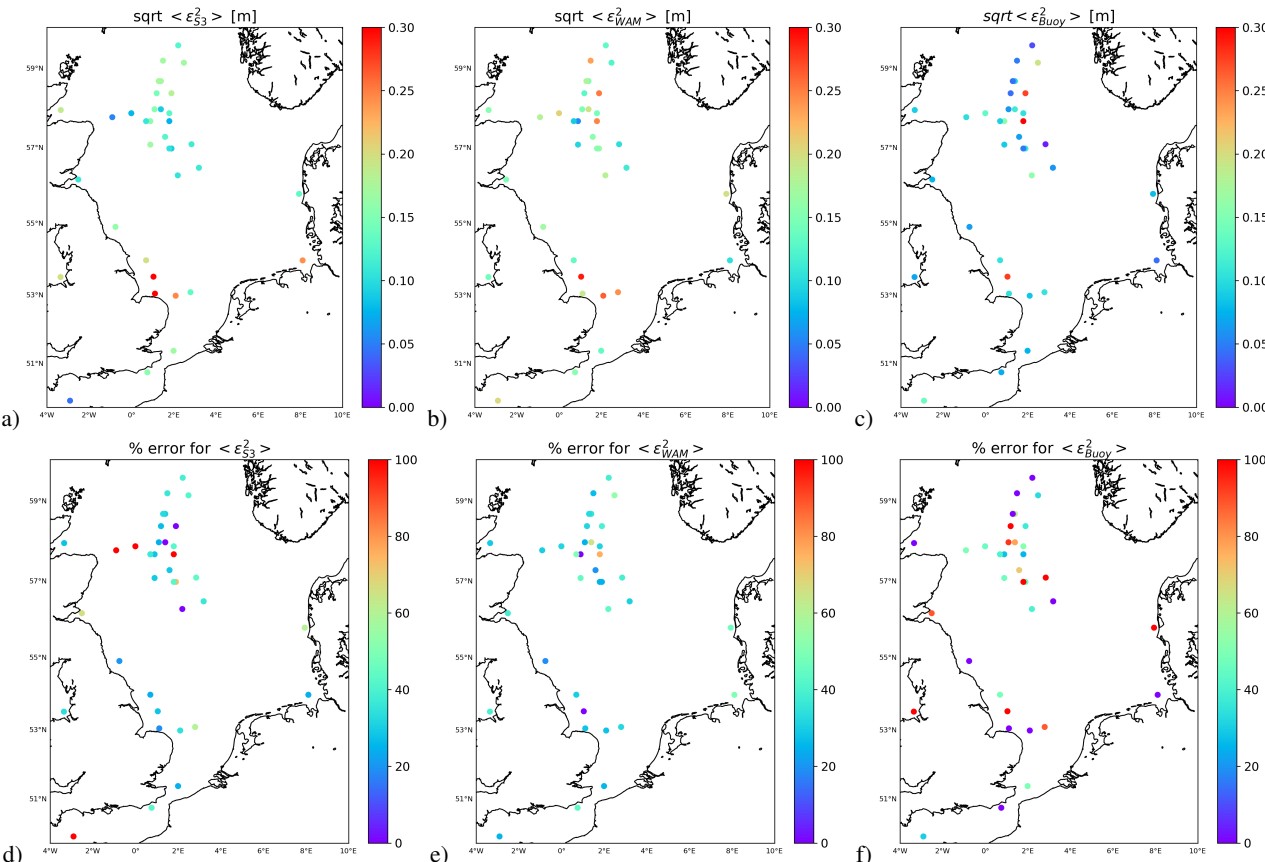

**Figure 7.** (a,b,c): Colour coded stochastic error standard deviations of wave heights provided by the SENTINEL-3a altimeter (a), the WAM model (b), and the insitu stations (c) estimated by *triple collocation*. (d,e,f): Relative uncertainties of the stochastic error estimates, derived by using the approach in Section 2.1.

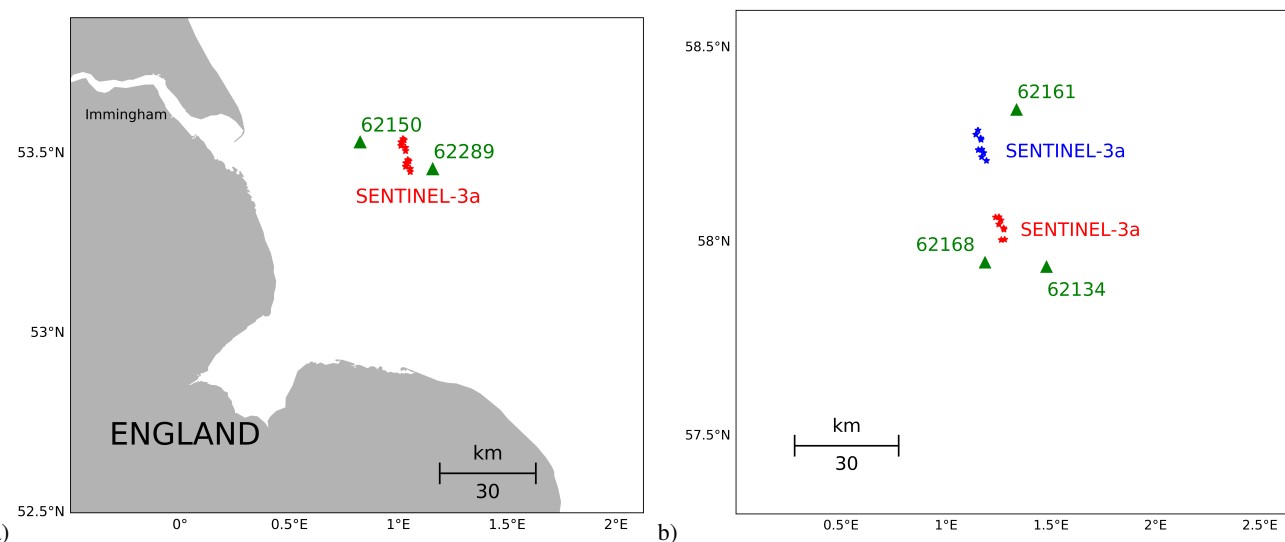

**Figure 8.** (a): Example of observation configuration with a descending SENTINEL-3a altimeter track passing between two insitu wave observation instruments ("62150" and "62289"), (b): Observation geometry with SENTINEL-3a track passing through a group of three insitu wave measurement devices. The blue and red altimeter measurements are used to estimate error correlations for both the altimeter and the numerical model.

*Competing interests.* The authors declare that they have no conflict of interest.

*Acknowledgements.* This publication has received funding from the European Union's H2020 Programme for Research, Technological Development and Demonstration under grant agreement no. H2020-EO-2016-730030-CEASELESS. We thank Jean Bidlot from ECMWF for providing GTS insitu data. BSH kindly gave access to waverider buoy measurements. We are greatful to Luciana Fenoglio-Marc from the University of Bonn for providing SENTINEL-3a altimeter data. We thank Arno Behrens from HZG for assistance with the wave model.

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
