# Peer review of "A multi collocation method for coastal zone observations with applications to SENTINEL-3a altimeter wave height data"

_Ocean Science, 2018_

## Referee Comment (RC1) · Anonymous Referee #1 · 22 Nov 2018

This paper describes a method to extend the established 'triple collocation' technique, used to quantify errors in measurement and forecast datasets, for use in the coastal zone and other regions where correlation lengthscales are short, and/or where observed data are sparse. This represents a significant addition to existing literature on triple collocation with some novel impacts. The paper is well written and clear and, as such, I would recommend it for publication subject to some minor corrections and additional clarifications/discussion points as outlined below.

Clarification / Discussion points:

[Figure]

The suggested additional discussion points are focused around the description of measured data in section 3 and the results in section 4.

Clarifications required in section 3 are as follows:

- (with reference to previous studies, e.g. Janssen et al., 2007) a choice has been made to use satellite altimeter data in its 1Hz form, whereas these data have previously been super-observed in order to match a representation scale close to that of model or in-situ data. There is no particular issue in using the data this way, but possibly this impacts some of the later results regarding error variability. So could the authors please clarify the representation scales attributed to each of the data sources?

- (as part of the above) the dataset from the JCOMM verification project supplies two versions of in-situ observations; the raw values, and a QC'd value at synoptic hours but derived from a mean of the waves over several hours surrounding this time. It is not entirely clear which of these was used (my impression is the former?) and what QC/super-observing procedures were applied to these and the BSH data.

- in addition to the offshore oil platforms (downward facing lasers/radars) and waveriders, one or two points in the JCOMM dataset are, I believe, measurements at lightvessels. It is worth noting that a known low bias exists in the reports from these locations, due to the hull response of the platform (Anderson, G., Carse, F., Saulter A., and J. Turton, 2016: Quantification of Bias of Wave Measurements from Lightvessels. J Op Oceanography http://dx.doi.org/10.1080/1755876X.2016.1239242)

Discussion points in section 4 are:

- the result that the buoys have the smallest errors is different to Janssen et al (2007)'s findings, in which buoy data were found to contain large errors. Caires and Sterl (2003) found something more in line with this study. This raises a question as to how much the results of triple collocation are influenced by the choice of in-situ data and use of super-observation. Janssen et al 'smoothed' their data significantly in attempting to

use a unified representation scale (needed for data assimilation) and then explained the result for in-situ data as due to significant variations in the way in-situ data was processed (subsequent papers, e.g. Durrant et al 2009, seem to confirm this). In this paper the authors appear to have used the data in a more raw form and, although different platforms make up the in-situ dataset, behaviour within a regional observation network may well be more self-consistent than the global dataset used by Janssen et al. For the purposes of this paper, it would therefore be useful for the authors to contextualise the treatment of the study data and results relative to some of these past studies. This is in order that readers can correctly attribute some of the headline results about buoy/altimeter errors to the choice of data processing rather than the updated triple collocation method.

- some expansion on the comments in the paragraph starting at P15-Line7 are, perhaps, warranted. For example, there is significant location to location variability in bias within buoy clusters in open waters (Figure 6), a number of buoys have high relative uncertainties (Figure 7), and one location in the southern North Sea shows similarly high stochastic errors to the two outliers identified in the northern North Sea. Combined with the known bias issues for lightvessels (some of which I think are included in this dataset) I think these results present an opportunity to ask whether in-situ networks, whilst a desirable reference, truly provide the consistency needed in this context?

Suggested minor corrections to text:

P1-Line10: 'presented method allows use of a large variety'

P1-Line12: 'sources is too big to assume that they'

P2-Line12: 'room for improvement, in particular'

P3-Line22: 'an estimation of cross covariance' (delete leading 'to')

P4-Line3: 'the track to assume that all three instruments'

P4-Line9: 'the spatial variation of the "truth" are required'

P4-Line16: 'with a small number of samples'

P4-Line21: 'This includes a new step in the analysis, in which estimation errors are quantified.'

P4-Line22: 'Section 5 describes the combination'

P5-Line6: 'the "truth" cannot, in general, be represented by'

P5-Line9: 'the approach in eq. 4 allows the addition of higher order terms'

Figure 5: 'Bias and calibration errors were corrected for the model and satellite'; I'm not sure I understand this statement in the context of the figure, please clarify or remove.

P15-Line28: 'allows estimation of the errors of all'

P17-Line1: 'allow an estimation of the uncertainties'

P17-Line10: 'allows an estimation of the errors'

P19-Line2: 'In this study we considered only linear models, but this is not a restriction of the method, since more sophisticated functional forms (e.g., bilinear functions) can be easily integrated.'; Is it worth commenting that such forms are likely to be required in near coastal zones, where nonlinear processes are more likely to drive the spatial variations than in the offshore?

P19-Line13: 'allowed a demonstration of the usefulness'

P19-Line16: 'biased high, in particular at higher sea states'
* * *

---

## Referee Comment (RC2) · Anonymous Referee #2 · 26 Nov 2018

The paper deals with relevant methodology to assess measurement and model errors when data are scattered in space. This is important, particularly to help validation of satellite data which due to its nature, it is difficult to obtain in-situ measurements precisely at same geographical location. The paper is presented in an organized manner, where first a standard approach (triple colocation method) is presented, then the extended method is shown and tested first with synthetic data and subsequently with real data. With the rapid increase of available data (in-situ measurements, satellites and models) this method is expected to be helpful on the assessment and identification

of error bars. For this reason, I consider the paper is worth of publishing considering some suggestions for discussions and improvements which would help readers to put the paper more in context. It seems that the authors want to give special focus on "coastal zone" as this is in the title, however the paper is missing more discussions about the method in the coastal zones, for example the implication of the assumptions for distance selected and the type of interpolation. Although the authors mention the heterogeneity of the coastal zone, probably this heterogeneity is not linear and interpolation methods might be difficult to apply if not considered the physical processes involved in the area where the different measurements come from. Within this context a discussion on what is the implication of the footprint of satellite for this method and in the coastal zone. This together with the performance during high sea states. A quantification of "high sea state" should also be given.

Specific comments: Line 17 page missing "s" in "in-situ wave observations" Line5 page2, please specify the time resolution of HF radar Line 3 page 3, as mention above, the direct application to coastal zone is not completely explored. Please specify what are the requirements considered when saying "special requirements of the coast in mind" Line 12, page 3, related to "the question about accuracy of error estimate. . . . . .Sentinel3a.." Is this solved in this paper? A short conclusion and recommendation should be added Line 15, page 3. Add "The interpolation of numerical model data to given observation locations is usually less critical if spatial resolution is appropriate" Lines 1:4, page 4. The assumption of linear combination might be not applicable in coastal zone. Line 12, page 4. The assumption of 10 km might be questionable and will have a strong impact in the coastal zone. As mentioned above more discussions would be beneficial Line 3, page 5. Please define variable "T" Line 5 page 7, Can you change the sentence to "Therefore the uncertainties of the estimated vector. . ."? Line 20, page 8, change "scaling factors" for "scaling parameters" to have a consistent nomenclature (see i.e. line 16 page 8) Line 12 page9, change "For the analysis is. . ." by "For the analysis in.." Line 24, page 9. Please mention the water depth of buoys Page 10, related to table 3. The table 3 is not clear. Please describe each

column in the table caption. Why first column appears as "stdv" as column 4, 5 and 6 and units are different. By looking at table 3 it should be easy to see the "truth errors" and also the ones obtained by the Monte Carlo simulation Line 6, page 11 Equation 41, and all the equations. Be sure all parameters are defined explicitly. Hs seems not defined. Line2-3 page 12, does this mean that satellite data are not "very" applicable for storm conditions near the coast? Please discuss Line 20, page 12. Please indicate if water level variations are considered in the wave model Line 23 page 13. Is "This is an important question" better as "This is and important result"? Line 30, page 13 Add "stochastic" before "error" to make it clearer. Same in line 26 of page 14 Line 26 page 15. Referring to "(red dots)" , please refer to corresponding figure Equation 50 and 51 use nomenclature (e.g. 62150) which should be introduced earlier, maybe in section of measurements if such specific naming convention is relevant. Line 7 page 17, introduce naming "north" and "south" to the locations Line 13 page 19. Replace "... was relatively small and allowed to.." by "... was relatively small, however it allowed ...." Line 16 page 19, please specify range of "higher sea states" and also with its relation to varying footprint and implications for coastal applications Caption of table 2 is missing the description of the mean (third column) Figure 3. Is it necessary to show 2 symbols in the legends of the subplots? Caption of figure 4. Mention that the red dashed line only indicates the zero

A references that is worth to consider to include: Kaighin A. McColl et al. (2014) Extended triple collocation: estimating errors and correlation coefficients with respect to an unknown target. Geophysical Research Letters.

---

## Author Comment (AC1) · 17 Jan 2019

**Response to reviewer's comments on the manuscript**

"A multi collocation method for coastal zone observations with applications to SENTINEL-3a altimeter wave height data"

by Johannes Schulz-Stellenfleth and Joanna Staneva

*We thank Reviewer 1 for many helpful and constructive comments. We appreciate the time and effort you have obviously invested in this. In the following, You find point by point responses to all comments given in the review. The original comments are given in bold black and the respective responses in green italic. The page and line numbers refer to the original version and do naturally not exactly match with the revised manuscript.*

**Anonymous Referee #1**

**This paper describes a method to extend the established 'triple collocation' technique, used to quantify errors in measurement and forecast datasets, for use in the coastal zone and other regions where correlation length scales are short, and/or where observed data are sparse. This represents a significant addition to existing literature on triple collocation with some novel impacts. The paper is well written and clear and, as such, I would recommend it for publication subject to some minor corrections and additional clarifications/discussion points as outlined below.**

**Clarification / Discussion points:**

**The suggested additional discussion points are focused around the description of measured data in section 3 and the results in section 4.**

**Clarifications required in section 3 are as follows:**

- **(with reference to previous studies, e.g. Janssen et al., 2007) a choice has been made to use satellite altimeter data in its 1Hz form, whereas these data have previously been super-observed in order to match a representation scale close to that of model or in-situ data. There is no particular issue in using the data this way, but possibly this impacts some of the later results regarding error variability. So could the authors please clarify the representation scales attributed to each of the data sources?**

  *Thanks for the comment – we have added the following text at the end of Section 3.3.*

  *"Compared to previous studies (Janssen et al, 2008; Caires and Sterl, 2003), the spatial resolutions of the three analysed data sources are in quite close agreement. The effective resolutions of the altimeter and the insitu instruments both depend on the actual sea state. For the altimeter typical footprint sizes are between 1 km and 10 km as explained in Section 3.1. For the insitu data, the translation of the typical 20 min averages to spatial averages is determined by the group velocity. For example, the energy propagates with about 15 km/hour, if the dominant wave length is 50 m long and the water is deep (>50 m). A 20 min temporal average would therefore correspond to a 5 km spatial average in this case, which is in good correspondence to the spatial model resolution of about 3.5 km. We have therefore used the original data for the analysis and not generated super-observations by averaging, as done in Janssen et al. (2008) and Caires and Sterl (2003), who used wave model data with significantly coarser resolution."*

- **(as part of the above) the dataset from the JCOMM verification project supplies two versions of in-situ observations; the raw values, and a QC'd value at synoptic hours but derived from a mean of the waves over several hours surrounding this time. It is not entirely clear which of these was used (my impression is the former?) and what QC/super-observing procedures were applied to these and the BSH data.**

  *The provided insitu data are hourly except for the BSH observations, which are every 30 min. These data are raw observations, which were run through basic sanity checks. We modified the last part of section 3.2 on page 12 according to:*

  *"The GTS data have a temporal sampling of 1 hour, while the BSH buoys provide observations every 30 minutes. The insitu observations represent raw values and were checked for unrealistic wave heights. Looking at all the insitu stations for the analysed period in summary, the provided significant wave heights were in the range between 0.1 m to 7.8 m. These are realistic values for the North Sea (Semedo et al., 2015) and hence all observations were used in the analysis.*

- **In addition to the offshore oil platforms (downward facing lasers/radars) and waveriders, one or two points in the JCOMM dataset are, I believe, measurements at lightvessels It is worth noting that a known low bias exists in the reports from these locations, due to the hull response of the platform (Anderson, G., Carse, F., Saulter A., and J. Turton, 2016: Quantification of Bias of Wave Measurements from Lightvessels. J Op Oceanography http://dx.doi.org/10.1080/1755876X.2016.1239242)**

  *We modified the text in Section 3.2. according to*

  *"Due to the lack of respective metadata, it was in general not possible to distinguish between different types of instruments, e.g., waverider buoys, lightships, or platform mounted devices. One exception is the station "62170" near the east English Channel entrance, which is identical to the light ship "F3" mentioned in Anderson et al.,(2016)."*

  *After eq. 42 we added the following text:*

  *"It is interesting to note, that for the location of the lightvessel ``62170" near the east entrance of the English Channel (see Fig.1) the satellite and the model show a positive bias of about 0.3 m and 0.2 m respectively. According to Anderson et al. (2016), one can expect a systematic low bias for wave height measurements from lightvessels of about 0.3 to 0.4 m. It is thus possible, that the estimated high bias for satellite and model is in this case an artefact caused by the violated assumption of bias free insitu observations."*

  *In addition, we added a label for the lightvessel "62170" in Fig. 1 and Fig. 2b.*

Discussion points in section 4 are:

- **the result that the buoys have the smallest errors is different to Janssen et al (2007)'s findings, in which buoy data were found to contain large errors. Caires and Sterl (2003) found something more in line with this study. This raises a question as to how much the results of triple collocation are influenced by the choice of in-situ data and use of super-observation. Janssen et al 'smoothed' their data significantly in attempting to use a unified representation scale (needed for data assimilation) and then explained the result for in-situ data as due to significant variations in the way in-situ data was processed (subsequent papers, e.g. Durrant et al 2009, seem to confirm this). In this paper the authors appear to have used the data in a more raw form and, although different platforms make up the in-situ dataset, behaviour within a regional observation network may well be more self-consistent than the global dataset used by Janssen et al. For the purposes of this paper, it would therefore be useful for the authors to contextualise the treatment of the study data and results relative to some of these past studies. This is in order that readers can correctly attribute some of the headline results about buoy/altimeter errors to the choice of data processing rather than the updated triple collocation method.**

  *Thanks for this comment. We have added the following text as a final paragraph in Section 4 on page 15:*

  *"The finding that, on average, the insitu stations have the smallest stochastic errors is at first sight in disagreement with results presented in Janssen et al. (2007). One has to take into account however, that there are a number of significant differences in the analysis. First of all, a global wave model with 55 km resolution was used in the former study, whereas the computational model grid used in our analysis has a resolution more than 10 times higher. It is unlikely however, that the coarser model resolution is the only factor, because Caires and Sterl (2003) also concluded, that the insitu stations have the smallest stochastic errors using wave model output with even coarser resolution (1.5$^o$) than used by Janssen et al. (2007). Both studies introduced altimeter super-observations (averages over subsequent measurements) to make the altimeter observations more consistent with the model estimates. In the present study this was not considered necessary, because the altimeter and model resolutions are in much closer agreement. The second major difference with respect to previous studies is the geographic locations and the type of altimeter data considered in the analysis. Janssen et al. (2007) investigated global ERS-2 and ENVISAT altimeter data sets, while Caires and Sterl (2003) concentrated on TOPEX and ERS-1 altimeter data acquired over the Pacific and the US east coast. This means that there are certainly differences both with regard to the background wave statistics and the satellite and insitu observation errors. A third important difference between the studies is the applied collocation criteria. Janssen et al. (2007) required the model, insitu and satellite estimates to be within 200 km distance and Caires and Sterl (2003) used a smaller collocation distance of 0.75$^o$ The allowed distance of 10 km used in the present study is still significantly smaller than that, and the collocation errors are therefore also likely to be smaller. For the above reasons one cannot conclude that the present study contradicts the results in Janssen et al. (2007). The conclusion is rather, that a common set of reference insitu data and collocation criteria are desirable to make different studies more comparable."*

- **some expansion on the comments in the paragraph starting at P15-Line7 are, perhaps, warranted. For example, there is significant location to location variability in bias within buoy clusters in open waters (Figure 6), a number of buoys have high relative uncertainties (Figure 7), and one location in the southern North Sea shows similarly high stochastic errors to the two outliers identified in the northern North Sea. Combined with the known bias issues for**

**lightvessels (some of which I think are included in this dataset) I think these results present an opportunity to ask whether in-situ networks, whilst a desirable reference, truly provide the consistency needed in this context?**

*We fully agree, that the heterogeneity of the insitu observations is a problem in the analysis. We have added the following text at the end of Section 4 to emphasize this point more: .*

*"It is evident that the observed heterogeneity of insitu measurements is a big complicating factor in the analysis. Wave model computations and satellite altimeter observations have reached a level of accuracy, where further improvements require a very careful selection and treatment of validation data sets. This in particular requires more knowledge about the type of insitu instruments and applied data processing techniques (e.g., averaging intervals). This could also be an argument for investments into dedicated validation instruments with more transparent and better documented error characteristics and quality control. The deployment of such instruments should take into account both research aspects and requirements for operational use."*

Suggested minor corrections to text:

**P1-Line10: 'presented method allows use of a large variety'**

*This was changed as suggested*

**P1-Line12: 'sources is too big to assume that they'**

*This was changed as suggested*

**P2-Line12: 'room for improvement, in particular'**

*This was corrected.*

**P3-Line22: 'an estimation of cross covariance' (delete leading 'to')**

*This was modified as suggested*

**P4-Line3: 'the track to assume that all three instruments'**

*Comma was removed*

**P4-Line9: 'the spatial variation of the "truth" are required'**

*Replaced "is" by "are"*

**P4-Line16: 'with a small number of samples'**

*Added "a"*

**P4-Line21: 'This includes a new step in the analysis, in which estimation errors are quantified.'**

*This was modified as suggested*

**P4-Line22: 'Section 5 describes the combination'**

*This was modified as suggested*

**P5-Line6: 'the "truth" cannot, in general, be represented by'**

This was modified as suggested

**P5-Line9: 'the approach in eq. 4 allows the addition of higher order terms'**

*This was modified as suggested*

**Figure 5: 'Bias and calibration errors were corrected for the model and satellite'; I'm not sure I understand this statement in the context of the figure, please clarify or remove.**

*We replaced this formulation by the following, which we hope is more clear:*

*"The model and satellite data sets were corrected according to the slope and bias parameters estimated in the collocation procedure.*

**P15-Line28: 'allows estimation of the errors of all'**

*This was corrected*

**P17-Line1: 'allow an estimation of the uncertainties'**

*This was modified as suggested*

**P17-Line10: 'allows an estimation of the errors'**

*This was adjusted accordingly*

**P19-Line2: 'In this study we considered only linear models, but this is not a restriction of the method, since more sophisticated functional forms (e.g., bilinear functions) can be easily integrated.'; Is it worth commenting that such forms are likely to be required in near coastal zones, where nonlinear processes are more likely to drive the spatial variations than in the offshore?**

*We agree and have added the following sentence at the end of the first paragraph of the conclusions on page 19:*

*"Such higher order approaches are certainly desirable for coastal areas with strong spatial variations, however they require a larger number of data sources (compare eq. 9)."*

**P19-Line13: 'allowed a demonstration of the usefulness'**

*Considering a comment from Reviewer 2, this part was reformulated as follows:*

*"The number of available samples was relatively small and estimation errors had therefore to be taken into account. The usefulness of the derived error bars for the interpretation of the data could be demonstrated."*

**P19-Line16: 'biased high, in particular at higher sea states'**

*This sentence was replaced following a comment by reviewer 2. .*

*Additional Changes*

- *Corrected the $n_o$ value for the 1D case in table 1 (from 6 to 5)*

- *Updated reference for Wiese et al, 2018*

- *Completed bibliography information (e.g., doi) for several references.*

**P19-Line16: 'biased high, in particular at higher sea states'**

*This sentence was replaced following a comment by reviewer 2. .*

---

## Author Comment (AC2) · 17 Jan 2019

**Response to reviewer's comments on the manuscript**

"A multi collocation method for coastal zone observations with applications to SENTINEL-3a altimeter wave height data"

by Johannes Schulz-Stellenfleth and Joanna Staneva

*We thank Reviewer 2 for many helpful and constructive comments. We appreciate the time you have obviously invested in this. In the following, You find point by point responses to all comments given in the review. The original comments are given in bold black and the respective responses in green italic. The page and line numbers refer to the original version and do naturally not exactly match with the revised manuscript.*

**Anonymous Referee #2**

**The paper deals with relevant methodology to assess measurement and model errors when data are scattered in space. This is important, particularly to help validation of satellite data which due to its nature, it is difficult to obtain in-situ measurements precisely at same geographical location. The paper is presented in an organized manner, where first a standard approach (triple colocation method) is presented, then the extended method is shown and tested first with synthetic data and subsequently with real data. With the rapid increase of available data (in-situ measurements, satellites and models) this method is expected to be helpful on the assessment and identification of error bars. For this reason, I consider the paper is worth of publishing considering some suggestions for discussions and improvements which would help readers to put the paper more in context.**

**It seems that the authors want to give special focus on "coastal zone" as this is in the title, however the paper is missing more discussions about the method in the coastal zones, for example the implication of the assumptions for distance selected and the type of interpolation. Although the authors mention the heterogeneity of the coastal zone, probably this heterogeneity is not linear and interpolation methods might be difficult to apply if not considered the physical processes involved in the area where the different measurements come from. Within this context a discussion on what is the implication of the footprint of satellite for this method and in the coastal zone. This together with the performance during high sea states. A quantification of "high sea state" should also be given.**

*We have added a new subsection 2.4, which discusses the implications of collocation distance and system resolutions based on the coastal background statistics presented in Section 2.3. This is supposed to put the focus on the special requirements of the coast, where we can usually expect stronger spatial gradients than in the open ocean. The consequences of such gradients for collocation errors and resolution related errors are discussed.*

*The expected errors for the triple collocation method are computed for different collocation distances in a coastal area with a wave height gradient (German Bight).*

*For the same area the subresolution wave height variance is estimated for different resolution cell sizes. It is explained, that this variance becomes part of the data set error in addition to pure instrumental errors.*

*The statement about the altimeter performance in "higher sea states" in the conclusions was reformulated and is hopefully more clear now.*

*We also hope that it is more clear now, that the linear interpolation method used in the presented analysis is of course not always realistic, but still a progress compared to the assumption of spatially constant wave heights, that have to be applied in the standard triple collocation method. It is also more emphasized now, that the multi-collocation method is not restricted to linear approaches, but that higher order interpolations require a larger number of data sources.*

Specific comments:

**Line 17 page missing "s" in "in-situ wave observations"**

*Replaced "observation" by "observations" in line 17 of the abstract.*

**Line5 page2, please specify the time resolution of HF radar**

*Replaced*

*"A few instruments, like HF radar are able to capture at least 2D surface currents with large coverage and high resolution quite nicely, but most instruments …"*

*by*

*"A few instruments, like HF radar are able to capture at least 2D surface currents with large coverage and high resolution quite nicely. Such systems have a typical range of about 100 km, spatial resolutions on a kilometre scale, and about 20 min sampling (Stanev et al., 2015). However, most instruments …*

**Line 3 page 3, as mention above, the direct application to coastal zone is not completely explored. Please specify what are the requirements considered when saying "special requirements of the coast in mind"**

*Thanks, we agree that this point should be explored in more detail. First of all, we have extended the text on page 3 as follows:*

*"In this study the triple collocation approach is extended  and adjusted with the special requirements of the coast in mind, where one can usually expect stronger gradients and smaller scale variations than in the open ocean."*

*Secondly, we hope that with the added Subsection 2.4. this issue is now presented with more clarity. The main point is that the stronger spatial gradients to be expected in near coastal areas have significant consequences both for collocation errors and errors, which are related to the averaging processes involved in the generation of the different data sets. In the new subsection this point is discussed first in more general terms and subsequently the theory is applied to the example of the German Bight. It is shown that the collocation errors for the standard triple collocation method can in fact be very significant. The multi-collocation method can reduce these errors at the cost of a larger required number of data sources.*

**Line 12, page 3, related to "the question about accuracy of error estimate: : :: : :Sentinel3a.." Is this solved in this paper? A short conclusion and recommendation should be added**

*We have added the following text in the conclusion around line 14 on page 19:*

*"For the analysed 16 months data set the estimation errors are significant, in particular if individual geographic locations are analysed. It would therefore be interesting to continue some parts of the analysis at a later stage of the SENTINEL-3a mission, when a larger data set will be available. More robust results are obtained, if averages over different spatially distributed insitu instruments are considered. ...."*

**Line 15, page 3. Add "The interpolation of numerical model data to given observation locations is usually less critical if spatial resolution is appropriate"**

*This was added as suggested.*

**Lines 1:4, page 4. The assumption of linear combination might be not applicable in coastal zone.**

*The multi-collocation method is not limited to linear interpolation approaches in general. However, if higher order approaches are used, a larger number of data sources is required. To make this point more clear we added*

*"We will concentrate on linear approximations in this study, however the method is able to deal with interpolation approaches of higher order, if a sufficient number of observations is available."*

*in line 10 on page 5.*

**Line 12, page 4. The assumption of 10 km might be questionable and will have a strong impact in the coastal zone. As mentioned above more discussions would be beneficial**

*In the added Subsection 2.4 an analysis is presented for the effects of the collocation distance on the triple collocation results in a coastal area with a spatial wave height gradient (German Bight). The analysis showed that there is a significant error increase going from 10 km allowed distance to 20 km allowed distance. In, general, the collocation distance is a compromise between the minimisation of collocation errors and the maximisation of the sample size. The 10 km distance limit used in this study is still smaller than the values used in previous studies ( Janssen et al.,2007; Caires and Sterl, 2003). If the collocation distance was reduced further, the smaller sample size would lead to increased estimation errors as described in Section 2.1.*

*We hope that with the added discussion in the new Subsection 2.4 this issue is more clear now.*

**Line 3, page 5. Please define variable "T"**

*We added*

*"Here and in the following, the symbol T denotes the transpose operation."*

*In line 4.*

**Line 5 page 7, Can you change the sentence to "Therefore the uncertainties of the estimated vector: : :"?**

*This was modified as suggested.*

**Line 20, page 8, change "scaling factors" for "scaling parameters" to have a consistent nomenclature (see i.e. line 16 page 8)**

*"scaling factors" was replaced by "scaling parameters" in line 20 on page 8, in line 16 on page 10, in line 13 page 11, in line 16 on page 14, and in the caption of table 4.*

**Line 12 page9, change "For the analysis is: : :" by "For the analysis in.."**

*Sorry, this was corrected.*

**Line 24, page 9. Please mention the water depth of buoys**

*We added information on the water depth for both buoys (25 m for "HEL" and 27 m for "ELB") in line 25 on page 9.*

**Page 10, related to table 3. The table 3 is not clear. Please describe each column in the table caption. Why first column appears as "stdv" as column 4, 5 and 6 and units are different. By looking at table 3 it should be easy to see the "truth errors" and also the ones obtained by the Monte Carlo simulation**

*Sorry, for the confusion – it is a little bit complicated, because the last columns refer to error standard deviations of estimates for variances and covariances, and therefore the units have to be $m^2$. As suggested, we reformulated the caption of table 3 explaining each of the columns in more detail.*

*"Parameters used for the Monte Carlo simulations in Section 2.4. The first two columns refer to the stochastic wave height error standard deviation (stdv) and variance (var) assumed for the considered data sources. The third column gives the assumed error cross covariance (covar) values for the two altimeter measurements and the two buoy data sets.The fourth column is the error standard deviation of the estimator for the observation error variances obtained by averaging over 1000 estimation experiments (AVEXP approach) . The values in column 5 refer to the same estimation errors, but derived by application of the method described in Section 2.1 (COMAT approach). The last column gives the COMAT and AVEXP standard deviations for the covariance estimation errors."*

**Line 6, page 11 Equation 41, and all the equations. Be sure all parameters are defined explicitly. Hs seems not defined.**

We added the definition of the symbol $H_s$ after equation 41.

**Line2-3 page 12, does this mean that satellite data are not "very" applicable for storm conditions near the coast? Please discuss**

*Unfortunately, the altimeter data processing, in particular near the coast, is very complicated (e.g., Chelton et al., 2001). Therefore, one has to be very careful with statements about the expected performance in certain conditions. Because we did not find previous publications , about the likely behaviour in high sea state conditions near the coast, and because the data analysed in this study are not sufficient to answer this question, we would prefer to avoid any statements that could mislead the reader. We have added the following text with an additional reference in the paragraph following eq. 41.*

*"In particular in coastal areas, the altimeter data processing is quite involved (Chelton et al., 2001), and a number of instrument and processing parameters can have a strong impact on the characteristics of the wave height estimates."*

*Ref: Chelton, Dudley B and Ries, John C and Haines, Bruce J and Fu, Lee-Lueng and Callahan, Philip S, Satellite altimetry, International geophysics,69,Elsevier,doi:10.1016/S0074-6142(01)80146-7 2001*

**Line 20, page 12. Please indicate if water level variations are considered in the wave model**

*We added the following sentence in the first paragraph of Section 3.3 to clarify this point:*

*"Spatial variations in bathymetry are taken into account, however temporal variations of water depth due to tides are not included in the simulations"*

**Line 28 page 13. Is "This is an important question" better as "This is and important result"?**

*This was changed as suggested*

**Line 30, page 13 Add "stochastic" before "error" to make it clearer. Same in line 26 of page 14**

*This was modified as suggested*

**Line 26 page 15. Referring to "(red dots)" , please refer to corresponding figure**

*We modified this to*

*"(red dots in Fig. 8a)"*

**Equation 50 and 51 use nomenclature (e.g. 62150) which should be introduced earlier, maybe in section of measurements if such specific naming convention is relevant.**

*We added the following sentence at the end of Section 3.2 on page 12:*

*"Some of the insitu stations shown in Fig.2b, which are referenced in the subsequent analysis, are labelled by either 5 digit numbers (e.g., ``62168''), or three character strings (e.g., ``ELB'')."*

**Line 7 page 17, introduce naming "north" and "south" to the locations**

*We reformulated this part as follows:*

*"Here, we concentrate on two locations covered by the satellite, which appear as two clusters in Fig. 8b. The "North'" group of satellite observations is shown in blue and the "South" group in red.*

**Line 13 page 19. Replace ": : : was relatively small and allowed to.." by ": : : was relatively small, however it allowed: : :."**

*The proposed formulation would change the meaning in a way we had not in mind in the original version. We hope that the following formulation makes the point a little bit more clear.*

*"The number of available samples was relatively small and estimation errors had therefore to be taken into account. The usefulness of the derived error bars for the interpretation of the data could be demonstrated."*

**Line 16 page 19, please specify range of "higher sea states" and also with its relation to varying footprint and implications for coastal applications**

*We agree, that this requires clarification.This comment is partly related to the comment about Line2-3 on page 12, where we tried to make it more clear, that the performance of the altimeter is depending on a larger number of system and processing parameter, which make statements about the expected performance in certain conditions very difficult.*

*We also noticed that confusion is caused by the formulation*

*" … slightly biased high, in particular at higher sea states."*

*because we have actually only estimated one bias value for the entire sea state range, and what we basically wanted to refer to, is the slope above unity observed for the satellite data on average (1.11). Together with the estimated slight high bias (0.07), this means, that the wave heights are overestimated by about 10% for above mean wave conditions. We have reformulated this part as follows:*

*"The analysis indicates, that on average the altimeter is overestimating wave heights by about 10% for above mean wave conditions."*

**Caption of table 2 is missing the description of the mean (third column)**

*We interchanged the second and third column of the table and added the missing information in the caption.*

**Figure 3. Is it necessary to show 2 symbols in the legends of the subplots?**

*We removed the first row of symbols in the legends of all subplots of Figure 3*

**Caption of figure 4. Mention that the red dashed line only indicates the zero**

*We added*

*" …in green, and the dashed red line indicating the zero position."*

*in the caption of figure 4.*

**A references that is worth to consider to include: Kaighin A. McColl et al. (2014) Extended triple collocation: estimating errors and correlation coefficients with respect to an unknown target. Geophysical Research Letters.**

*Thanks - we added this reference in the paragraph following equation 3.*

*Additional Changes*

- *Corrected the $n_o$ value for the 1D case in table 1 (from 6 to 5)*

- *Updated reference for Wiese et al, 2018*

- *Completed bibliography information (e.g., doi) for several references.*